# Teaching Pretrained Language Models to Think Deeper with Retrofitted Recurrence

## Abstract

Recent advances in depth-recurrent language models show that recurrence can decouple train-time compute and parameter count from test-time compute. In this work, we study how to retrofit existing pretrained non-recurrent language models into depth-recurrent models. We find that using a curriculum of recurrences to increase the effective depth of the model over the course of training preserves performance while reducing total computational cost. In our experiments on grade-school math, we train on math data from Common Crawl and observe that retrofitting pretrained models to be depth-recurrent results in better performance at a given training compute budget than simply post-training the original non-recurrent language model. Further, we train our retrofitted recurrent models on a mixture of FineWeb-Edu and high-quality Nemotron general and math data. We observe that retrofitting can yield performant, general-purpose depth-recurrent language models that improve with test-time compute via scaled recurrence, outperforming static-depth post-trained baselines on a range of common benchmarks including: GSM8K, ARC and PIQA.

## 1 Introduction

*Test-time compute scaling* refers to the use of additional computation during inference to improve model outputs. By decoupling computation intensity from model size, test-time compute scaling achieves superior benchmark scores without requiring more model parameters or additional pretraining. The mainstream paradigm for scaling test-time compute involves generating many tokens, either in chain-of-thought traces or by generating many candidate solutions and choosing the best (Snell et al., 2024; Guo et al., 2025). An emerging alternative paradigm for test-time scaling leverages depth-recurrence, by which a language model can simply recur layers for more iterations to expend more compute. Depth-recurrence has the advantage that increasing compute does not increase memory consumption or context size during inference. Moreover, not requiring the model to verbalize thoughts as tokens may allow for more complex reasoning to happen within the latent space where there is higher information bandwidth. Finally, recurrent networks can be trained on standard data sources and do not require training with bespoke reasoning traces in the domain of interest.

Geiping et al. (2025) pretrain a depth-recurrent transformer from scratch on 800 billion tokens at substantial cost. Although their model can reuse parameters at test time to scale up compute and improve performance, their work also uses a large number of recurrent iterations during training, which significantly slows down training compared to a feedforward model with the same parameter count. This inspires us to focus on the training efficiency of depth-recurrent models.

In this work, we study fast procedures for retrofitting feedforward models to be depth-recurrent models through continued pretraining. Because transformer models include residual connections (He et al., 2015) that write updates back into the same residual stream, transformer layers operate in a shared representation space (Elhage et al., 2021). This makes it possible to "loop" a block of layers from a pretrained LLM by feeding the output of the block back into itself as input. By training a model while it operates in this looped mode, the model learns to exploit recurrence to improve performance. Our main experiments demonstrate that TinyLlama-1.1B-intermediate-step-1431k-3T (Zhang et al., 2024b) and Llama-3.2-1B (Grattafiori et al., 2024) can be retrofitted into recurrent depth transformers. We observe that doing so improves performance on reasoning tasks that are

known to differentially benefit from additional test-time compute (Geiping et al., 2025). Moreover, with a well formed data mix we train performant general purpose depth-recurrent language models. Our model is competitive with, and in some achieves higher accuracy than, Huginn-0125 (Geiping et al., 2025) with approximately a quarter of the parameters..

Our central goal is to train highly performing depth-recurrent models in the most compute efficient way possible. There are two success metrics we identify:

**(1) Improving over Random Initialization.** We want the retrofitted model, initialized from pretrained weights, to outperform a model trained from random initialization on a per-training-FLOP basis. Since parameters are both added to and removed from the original model when retrofitting to be depth-recurrent, this knowledge transfer goal is non-trivial. We show in Figure 2 that initializing a depth-recurrent model from Llama-3.2-1B weights strongly outperforms the randomly initialized model in terms of loss and benchmark accuracy per training FLOP.

**(2) Improving over Fixed Depth Models.** We want the performance of the pretrained model to increase after retrofitting. We find that with a well-formed data curriculum, recurrence results in an increase in GSM8K accuracy while maintaining high accuracy on a broad suite of language modeling benchmarks (see Figure 7 and Table 1).

Overall, we show that retrofitting recurrence into pretrained language models is an efficient way to train depth-recurrent models that punch above their compute class. In summary, our contributions are as follows:

1. We show that initializing parameters of recurrent models from those of a pretrained feedforward model is significantly more efficient than using a random initialization (Figure 2).

2. We propose a curriculum over recurrent depths, slowly increasing the average number of recurrent iterations during training to maintain performance while improving training speed (Figure 3).

3. We show that, using Common Crawl math data, we can retrofit TinyLlama and Llama-3.2 models into recurrent models that achieve better GSM8K performance than base models (Figures 5 and 6).

4. While we remove layers when retrofitting feedforward models to recurrent ones, we find that introducing a "healing" period with minimal distribution shift allows us to recover basic language modeling performance before switching to task-specific data to further refine the depth-recurrent model's reasoning performance whist maintaining general language modeling performance (Figure 7 and Table 1).

## 2 RELATED WORK

**Recurrent models.** It has been shown that "universal transformers" based on recurrence are Turing-complete (Dehghani et al., 2018). Recurrent transformers with weight shared layers but a fixed layer repetition count have been studied in detail (Lan et al., 2019; Takase & Kiyono, 2021; Fan et al., 2024; Bae et al., 2024; Gao et al., 2024; Ng & Wang, 2024; Csordás et al., 2024; McLeish et al., 2024; Saunshi et al., 2025; Zeng et al., 2025). Adaptive-depth mechanisms have been studied with the specific goal of increasing computation efficiency (Graves, 2016; Elbayad et al., 2019; Schwarzschild et al., 2021; Bansal et al., 2022). A more advanced class of recurrent transformer can utilize an internal mechanism to exit after a data-dependent number of recurrences (Geiping et al., 2025; Aleksandrov et al., 2025; Chen et al., 2025; Bae et al., 2025). Raposo et al. (2024) propose mixture of depths models which adaptively route tokens through or around each transformer block. Mohtashami et al. (2023) augment mixture of depths with weight sharing, extended by Bae et al. (2025) with adaptive exiting to further increase efficiency.

**Model surgery.** There is a rich literature on methods for making post-hoc changes to model architecture and size (Chen et al., 2015; Wei et al., 2016). Relatedly, early exiting (Panda et al., 2016; Elbayad et al., 2019), has been used to speed up inference by skipping layers and using speculative decoding (Del Corro et al., 2023; Elhoushi et al., 2024; Liu et al., 2024). Moreover, Ye et al. (2025) explore converting a pretrained causal language model to a diffusion language model by removing the causal mask. Li et al. (2025) finetune looped models initialized from the GPT-2 (Brown et al., 2020) and OPT (Zhang et al., 2022) checkpoints finding small gains from finetuning and looping under-trained models on multiple choice benchmarks over the base checkpoints. Most related to ours is the work of Bae et al. (2024) which studies converting pretrained transformer language models into recurrent models using just 2 or 3 recursions. Notably, the authors maintain the same shape as

the base model and require low rank adapters (Hu et al., 2022) to recover performance of the base model. This means that the models cannot adaptively expend compute at test time by recurring for longer. Notably, Bae et al. (2024) find that recurring more leads to larger performance decreases in the post-trained model. Unlike Bae et al. (2024), our approach does not require distillation or auxiliary adapters. We focus on over-trained transformers and recur them for many iterations with adaptive recurrence.

**Latent reasoning.** Wang et al. (2025) introduce the hierarchical reasoning model (HRM); an architecture designed to better align with certain anthropomorphic biases for compositional intelligence. However, ARC Prize Team (2025) performs further ablations on the HRM architecture and finds only the main recurrence is needed for reasoning performance, reducing the HRM to a model similar to that of Geiping et al. (2025) without the ability to extrapolate in recurrence. We begin our own research by re-purposing aspects of the pretraining recipe developed by Geiping et al. (2025) to train a large recurrent language model from scratch; the first work to establish that latent reasoning as a scalable, alternate approach for pretraining transformer language models. We detail how our architecture and training recipe is derived from theirs more formally in Section 3.

We provide an extended discussion of other related work in Appendix A.

## 3 ARCHITECTURE & INITIALIZATION

This section describes the exact experimental setup we use to study depth-recurrence in decoder-only transformer language models, before we describe our results from training in Section 4. Throughout this work we use "initialize" to describe what the weights of the model at step 0 are based on, and "retrofit" to describe the process of training the looped models from a pretrained models weights.

**Model Definition.** Using the same notation as Geiping et al. (2025), here we define the structure of the class of recurrent models we study. We define $P$ as the prelude, $R$ as the recurrent block and $C$ as the coda; each of which is a set of unique transformer blocks with the embeddings included in $P$ and unembeddings in $C$. Given vocabulary set $V$, for an input sequence $\mathbf{x} \in V^n$ and a number of recurrences $r$, the model output distribution $\mathbf{p}$ is defined as follows.

$$\begin{array}{ccc} Prelude & Recurrent\ Block & Coda \\ \mathbf{e} = P(\mathbf{x}) & \mathbf{s}_0 \sim \mathcal{N}(\mathbf{0}, \sigma^2 I_n),\ \ \mathbf{s}_i = R(\mathbf{e}, \mathbf{s}_{i-1})\ \text{ for }\ i \in \{1, \ldots, r\} & \mathbf{p} = C(\mathbf{s}_r) \end{array}$$

Geiping et al. (2025) use a "scalable initialization" (Takase et al., 2023) for their Huginn-0125 model. Such schemes allow model shape to be altered whilst maintaining training stability. We also use this random initialization when training from scratch. To allow for adaptive recurrence at test time, Geiping et al. (2025) sample $r$ from a Poisson-Lognormal distribution with a mean of 32 at each training step. They also employ a truncated backpropagation procedure, only propagating gradients through at most the last 8 passes through $R$. This reduces training time and allows for very large values of $r$ without exhausting GPU memory.

We follow the Llama model conventions (Grattafiori et al., 2024; Zhang et al., 2024b) which differ from those used by Geiping et al. (2025). Specifically, Geiping et al. (2025) use normalizations four times in each decoder block and additionally use the final norm before the coda; we reduce to two norms in each decoder block and remove the dual use of the final layer norm. We also use grouped-query attention (Ainslie et al., 2023), train all models with a context length of 1024, and do not weight-tie the embedding and unembedding layers. We present additional technical training details in Appendix B.

**Retrofitting Recurrence.** Similar to Geiping et al. (2025), we use tuple notation to define the number of transformer layers in each of the prelude, recurrent block, and coda. For example, $(2, 4, 2)$ means there are 2 transformer layers in the prelude, 4 in the recurrent block, and 2 in the coda. To reduce latency at large numbers of test recurrences, we do not use every layer from the pretrained model when adapting it into a depth-recurrent model. We find that selecting the early layers for the prelude and later layers for the recurrent block and coda performs best (see Figures 1 and 11). For example, if the model we are using has 22 layers and we take a $(4, 8, 4)$ configuration. This corresponds to selecting layers $[0, 1, 2, 3], [10, 11, 12, 13, 14, 15, 16, 17], [18, 19, 20, 21]$; we use this selection for our $(4, 8, 4)$ TinyLlama based models. We detail the exact parameter counts and layers taken from pretrained models for our recurrent models in Appendix E. In Appendix Figure 11, we compare

to the ShortGPT pruning method (Men et al., 2024) to select layers to drop from the parent model when forming the recurrent model. We find our selection to be better for depth-recurrent model post-training. We also compare to taking all layers from TinyLlama to form a $(6, 10, 6)$ model and to a $(2, 4, 2)$ TinyLlama model in Appendix Figure 12.

We emphasize that although the two models we analyze in this paper share the "llama" name they are different models, trained independently. The two models are different shapes, with TinyLlama being 6 layers deeper than Llama-3.2, but narrower (smaller residual stream) as they both contain approximately 1 billion parameters. TinyLlama uses the Llama-2 vocabulary, whereas Llama-3.2 uses a vocabulary over $4\times$ larger. Finally, TinyLlama is trained with a next token prediction cross entropy loss from random initialization for 3 trillion tokens, whereas Llama-3.2 is initialized by pruning Llama-3.1-8B and then using logit level distillation from Llama-3.1-8B and Llama-3.1-70B for 9 trillion tokens (Meta, 2024).

**Calculating Training Cost.** For a recurrent model, the number of *unique parameters* refers to the number of distinct, trainable parameters in the model without double counting parameters that are shared across recurrences; we simply use the term *parameters* in this paper[1]. One can also consider the *effective parameters* of a recurrent model by including repetitions across recurrences. However, for clarity, throughout the rest of the work we quantify the size of a recurrent model evaluated at different depths in terms of Floating Point Operations (*FLOPs*) rather than describing parameter re-use. In other words, increasing the number of iterations performed by the recurrent block increases the amount of computation invested while number of actual parameters in the model remains fixed.

When calculating training FLOPs for standard fixed-depth transformers, we use the approximation FLOPs $= 6ND$ (Kaplan et al., 2020), where $N$ is non-embedding parameters and $D$ is number of training tokens. This approximation arises because on the forward pass 2 floating point operations are computed per parameter and on the backward pass 4 floating point operations are computed per parameter. Hence is a parameter is used without gradients only 2 floating point operations are required. However, recurrent models require a different rule. As we only backpropagate through at most the last 8 iterations of the recurrent block, we split the *effective parameter* count ($N$) into two parts: $N_1$ which includes all parameters with gradients recorded and $N_2$ which includes all parameters that are used in the forward pass without gradients. We calculate $N_1$ and $N_2$ using the mean number of recurrences during training. This gives $FLOPs = (6N_1 + 2N_2)D$ for our recurrent models.

## 4 TRAINING RECURRENT LANGUAGE MODELS

Our main experimental results are presented in four subsections, focusing on achieving our goals set out in Section 1.. In Section 4.1, we find that a pretrained initialization outperforms a random initialization in terms of loss and benchmark performance. Then, in Section 4.2, we use a curriculum to schedule the mean of the Poisson-Lognormal distribution, showing that this can reduce training costs without negatively impacting loss. In Section 4.3, we show that depth-recurrent post-training is more efficient than training non-recurrent models for grade school math problems. Finally, in Section 4.4, we demonstrate that with a good data curriculum, depth-recurrent models can be good general language models in addition to achieving higher accuracy on grade school math problems despite having fewer parameters.

### 4.1 EFFICIENTLY INITIALIZING RECURRENT TRANSFORMERS

We begin by demonstrating that **using a pretrained initialization outperforms a random initialization for depth-recurrent models**. In Figure 1, we perform a search over which layers to select when forming a recurrent model and removing layers. We find taking earlier layers to form the prelude and later layers to form the recurrent block and coda lead to the lowest training loss. Hence, throughout the paper we select these layers when retrofitting recurrence. In Figure 11, we find current layer removal strategies are suboptimal for the objective of retrofitting recurrence.

Following this search, we train two models for approximately 120 billion tokens on FineWeb-Edu (Penedo et al., 2024) data with a mean number of recurrences of 32. Figure 2 visualizes the

---

[1]We also exclude embedding and unembedding parameters in this count.

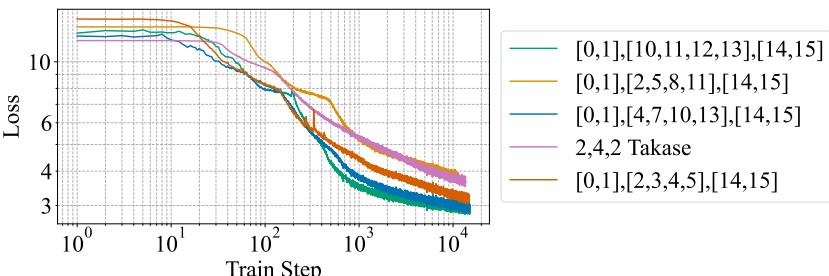

Figure 1: **Initializing the prelude from early layers, and the recurrent block and coda from later layers of Llama-3.2 gives the best training loss.** We measure the training loss on Fineweb-Edu with different layer selections from the depth 16 Llama-3.2. We find taking early layers for the prelude, and later layers for the recurrent block and coda to be best.

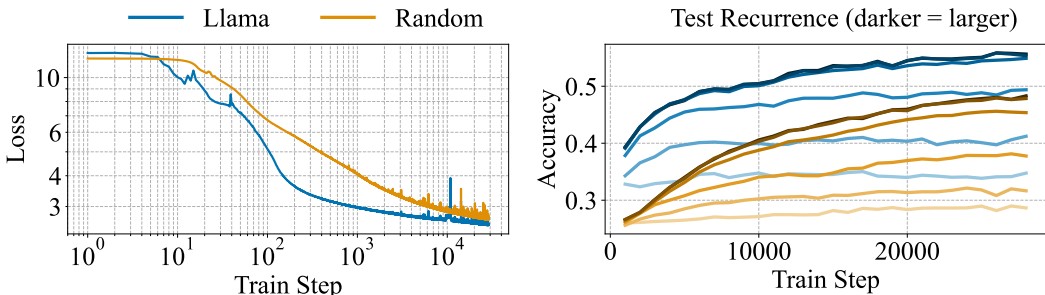

Figure 2: **Initializing from pretrained Llama-3.2 layers gives a significant advantage in loss and benchmark accuracy. Left:** Loss over training step for 120 billion tokens for models initialized from Llama-3.2-1B layers and randomly (Takase et al., 2023). Although starting higher, the model initialized from Llama-3.2 weights achieves lower losses consistently than the model initialized randomly. **Right:** Zero shot accuracy on Hellaswag (Zellers et al., 2019) over training step for recurrences $[1, 2, 4, 8, 16, 32]$. We see the Llama-3.2 based model (blue) achieves higher accuracy quicker and leverages recurrence effectively from early training steps. We record accuracy over recurrence for a suite of language modeling benchmarks in Appendix Table 2.

training loss and Hellaswag accuracy curves over training for a $(2, 4, 2)$ model initialized from `Llama-3.2-1B` and from random initialization, following Takase et al. (2023). On the left, we see the initialization from pretrained Llama-3.2 layers yields a large efficiency gain in terms of loss. On the right, we show that the model initialized from pretrained Llama-3.2 layers achieves higher benchmark accuracy earlier on Hellaswag (Zellers et al., 2019). By training step 1000, the Llama-3.2 initialized model is already leveraging recurrence to increase accuracy, unlike the random initialization for which all recurrences are achieving just over random accuracy.

In Appendix Table 2, we show the accuracy at $28,000$ steps for both models over multiple recurrence levels on a suite of language modeling benchmarks, finding that **initializing from pretrained Llama-3.2 weights causes a significant increase in accuracy in all cases**. In Appendix C.1, we also present additional experiments including a cooldown for 12 billion additional tokens. Extrapolating the loss curves in log-linear space suggests it would take at least approximately 950 billion tokens for these loss curves to intersect (see Appendix Figure 8). We emphasize that this number is likely an underestimate of the true number of tokens required for the models to achieve the same loss as the curves are still not perfectly log-linear at the end of our actual data.

## 4.2 SCHEDULING RECURRENCES

Using truncated backpropagation means the forward pass for our recurrent models consumes a larger share of runtime than it would for a non-recurrent model. Hence, reducing the time spent

on the forward pass for our models has a large impact on training time. With this insight, we explore an efficient curriculum which schedules the mean of the Poisson-Lognormal distribution. This curriculum is analogous to the gradual stacking technique (Gong et al., 2019; Reddi et al., 2023; Saunshi et al., 2024; Du et al., 2024) which increases the depth of a non-recurrent model by duplicating layers within the model during training and then training them independently. We visualize our curriculum in Appendix Figure 13.

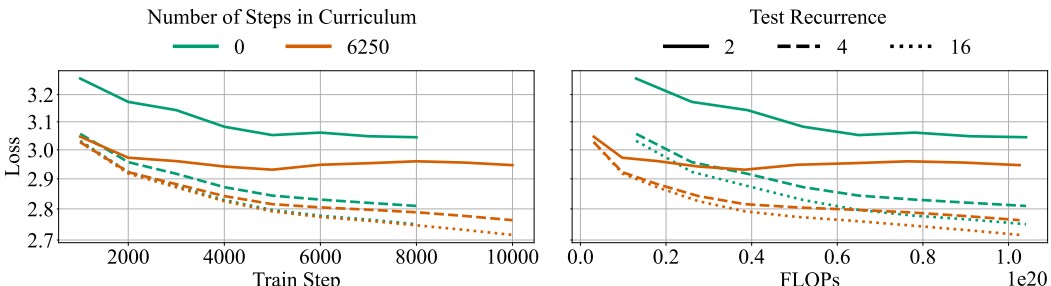

Figure 3: **Scheduling the mean of the depth distribution is efficient in terms of *both* data and compute.** We report validation loss over multiple recurrent depths in terms on steps on the left and in terms of FLOPs on the right. We see that scheduling the number of recurrences up to the final mean (32) over a long period of training decreases the validation loss, hence the curriculum is both data and compute efficient. Alternative length curricula and more test recurrent depths are shown in Appendix Figure 14.

Figure 3 measures the validation loss on one million tokens taken every 1000 training steps for $(2, 4, 2)$ models initialized from Llama-3.2 layers. This is the same as in Figure 2 but for a shorter time horizon of 48 hours on 4 MI300A GPUs which equates to approximately $2e^{20}$ FLOPs. In Figure 3 (left), we see that scheduling the recurrent depth has a small positive impact on the validation loss as a function of steps. Furthermore, on the right, we see that **scheduling greatly improves the efficiency in terms of loss improvement as a function of FLOPs spent during training**. In all subsequent experiments we use a curriculum for a conservative $25\%$ of total steps. In Appendix C.2, we show that scheduling the maximum backpropagation depth over training is better in terms of FLOPs but worse in terms of steps, hence less efficient than scheduling the mean depth but still valuable when trying to reach the lowest possible loss in a period of time. We also investigate using curricula when training from scratch in Appendix Figure 16.

### 4.3 How to Retrofit Recurrence

Next, we investigate how to efficiently retrofit depth-recurrence into pretrained non-recurrent transformers. First, we find Muon to be a better optimizer than AdamW when training recurrent models in Section 4.3.1. In Section 4.3.2, we analyze TinyLlama and Llama-3.2 models. In both cases, under the same training FLOP budget, depth-recurrent models with fewer parameters can achieve higher accuracy on grade school math problems than the non-recurrent parent model.

### 4.3.1 Optimization

We begin by initializing models from `TinyLlama-1.1B-intermediate-step-1431k-3T`. We consider $(4, 8, 4)$ TinyLlama models, dropping out 6 layers (layers $4, 5, 6, 7, 8$ and $9$, using 0-indexing) from the original model. This yields approximately 700 million remaining parameters in our recurrent models, $72.7\%$ of the parameters in the non-recurrent TinyLlama. That is to say, the depth-recurrent models have significantly fewer parameters than their non-recurrent parent models. Full parameter counts are provided in Appendix E.

In Figure 4 (left), **the Muon optimizer improves over AdamW** for our recurrent models as it achieves lower loss and removes loss spikes during training. For the non-recurrent TinyLlama models, the difference is much less pronounced, but we still see a small gain using the Muon optimizer. We smooth the loss over 50 steps to make this more visible in the plot. In Figure 4, we also compare to the variant of AdamW which is used by Geiping et al. (2025), and we refer to this variant as AdamW*.

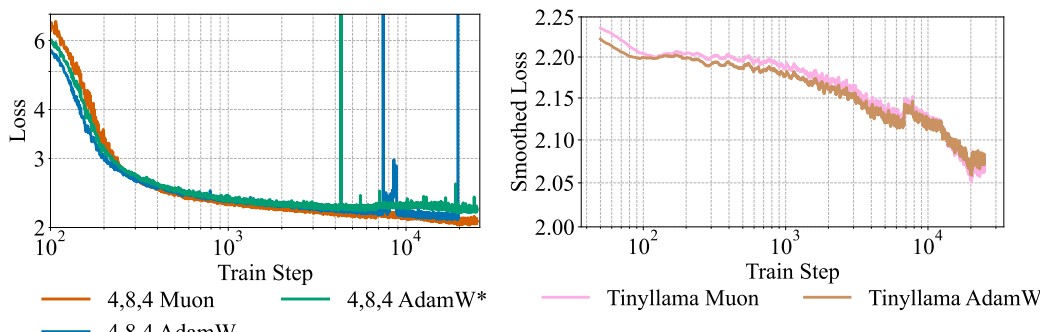

Figure 4: **Muon improves over AdamW when training recurrent models. Left:** Loss vs. step for multiple training runs on the same data order with different optimizers, using a learning rate of $5e^{-5}$ for AdamW and $0.001$ for Muon. Muon is the most stable and achieves the lowest loss for recurrent models. Note, the AdamW line ends early as the loss spikes and becomes NaN. **Right:** Loss (smoothed over $50$ steps) vs. step for AdamW and Muon. For the non-recurrent TinyLlama model there is minimal difference between optimizers.

AdamW* modifies AdamW by including update clipping, removing the $\varepsilon$ constant (Wortsman et al., 2023; Everett et al., 2024), and using a different decoupling method than the PyTorch AdamW implementation (Schaipp, 2024). In subsequent experiments, we optimize all models with Muon.

### 4.3.2 RECURRENT MODELS ARE EFFICIENT TO TRAIN

In our next set of experiments, while we continue training our $(4, 8, 4)$ TinyLlama models, we build another set of models initialized from the weights of Llama-3.2-1B. For Llama-3.2, we construct $(4, 6, 4)$ configurations, removing 2 layers (layers 4 and 5 with 0 indexing) from the pretrained model. This leaves approximately $850$ million remaining parameters in the recurrent model, which equates to $87.5\%$ of the pretrained models parameters. As in experiments with TinyLlama, this means the depth-recurrent Llama-3.2 models also have many fewer parameters than their non-recurrent parent models.

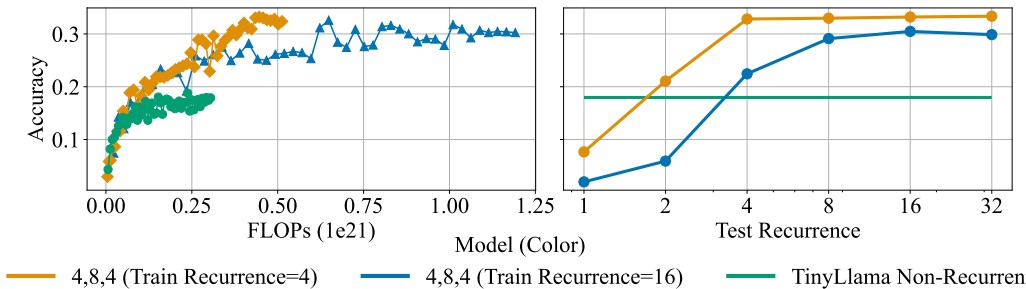

Figure 5: **Recurrence improves reasoning on GSM8K for TinyLlama, even when controlling for FLOPs.** We train $(4, 8, 4)$ and non-recurrent models for approximately 50 billion tokens of Nemotron-CC-Math-v1 data. **Left:** We plot accuracy over the number of FLOPs used during training. We see that recurrent models can efficiently outperform the non-recurrent baseline. **Right:** We plot accuracy over the number of recurrences for inference. We see the recurrent models are competitive with the fixed depth baseline and can outperform it by using more recurrences and therefore more FLOPs. We plot each individual models accuracy over training step and recurrence in full in Appendix C.3.1, including for training recurrence 8 and 32. Evaluations on the final checkpoint over tasks shown in Table 1 are in Appendix Table 3. We note a gain of up to $3\%$ on MMLU for the recurrent models over the non-recurrent baseline.

In Figures 5 and 6, we train models on approximately 50 billion tokens of Nemotron-CC-Math-v1-4plus (Mahabadi et al., 2025) data and evaluate on GSM8K (Cobbe et al., 2021), for TinyLlama and Llama-3.2 respectively. For our GSM8K evaluations, we

use a single shot example in context when evaluating, recording the flexible extract accuracy to avoid formatting of the answer being a confounder. Controlling for training FLOPs, both Figures 5 and 6 (left) show that it is highly efficient to train recurrent models. The depth-recurrent models achieve comparable performance to the non-recurrent baseline when evaluated at smaller training budgets but continue to improve as more compute is invested while accuracy for the non-recurrent model plateaus. We emphasize that all of these experiments utilize the same training dataset presented in the same order. The differences in curve length come from the additional FLOPs required to train the recurrent models (which require more FLOPs per parameter) for the same number of steps. The end of each line shown in Figures 5 and 6 (left) corresponds to the exact same number of tokens seen for each model.

In Figures 5 and 6 (right), we plot accuracy against number of recurrences used during inference for the models at the end of training. We see that recurrent models improve performance over the non-recurrent baseline significantly when utilizing more test-time compute. Moreover, combining this with Appendix Figures 18, 22, and Appendix C.3.4, we conclude that recurrent models are competitive on a per-FLOP basis for inference despite containing fewer trainable parameters at any FLOPs count. Overall, **depth-recurrent models are able to leverage compute to achieve a higher overall performance with fewer parameters than their non-recurrent counterparts**.

While our results for TinyLlama and Llama-3.2 are generally congruous, we do note some key differences in the results. We see that the best model in terms of training recurrences differs between the two models (4 for TinyLlama, 16 for Llama-3.2), and we hypothesize that this discrepancy is due to the different model shapes and pretraining biases. Furthermore, the accuracy achieved by the Llama-3.2 recurrent model in Figure 6 is higher than that of the one based on TinyLlama in Figure 5 which corresponds with a similar performance difference between the two non-recurrent baselines. This suggests that **using a stronger set of pretrained weights transfers additional knowledge to the final depth-recurrent model.**

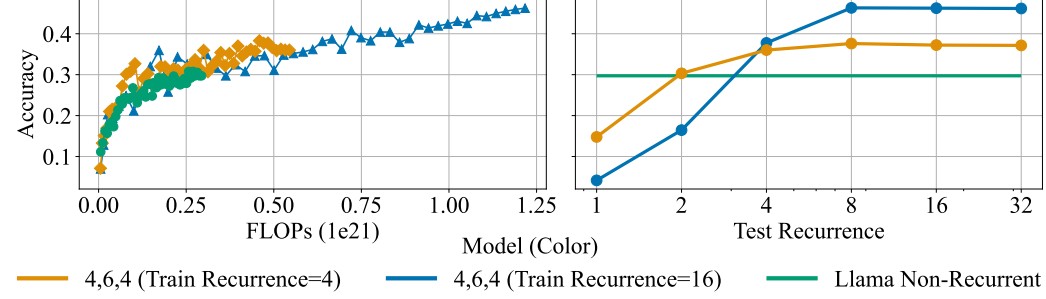

Figure 6: **Recurrence efficiently improves reasoning for Llama-3.2. Left:** We plot accuracy over the number of FLOPs used during training. We see that recurrent models can efficiently outperform the non-recurrent baseline when trained on the same tokens. **Right:** We plot accuracy over the number of recurrences used for inference. We see the recurrent models are competitive with the fixed depth baseline (green horizontal line) and can outperform it by using more recurrences and therefore more FLOPs. We plot each individual models accuracy over training and recurrence in full in Appendix C.3.2, including for training recurrence 8 and 32. Evaluations on the final checkpoint over tasks shown in Table 1 are in Appendix Table 4.

## 4.4 DATA MIXTURES

In previous experiments, we observe that training strictly on math data slightly degrades performance on non-reasoning based evaluations such as Hellaswag and OpenbookQA (see Appendix Tables 3 and 4). To address this degradation, we train on a mix of FineWeb-Edu (Penedo et al., 2024), Nemotron-Pretraining-SFT-v1-General (NVIDIA et al., 2025), and Nemotron-Pretraining-SFT-v1-Math (NVIDIA et al., 2025). We remove all data generated by reinforcement-learned reasoning models such as Deepseek-R1 and the user-assistant tags from the Nemotron-Pretraining-SFT-v1 data when training.

In our first experiment on data mixtures we train $(4, 8, 4)$ TinyLlama models for 26 billion tokens on an even mix of the three datasets; we call this *single phase* training. Since we remove

layers during recurrent retrofitting, we hypothesize that the depth-recurrent models must first recover their basic language modeling abilities before they can efficiently learn the high-quality Nemotron-Pretraining-SFT-v1 data. To test this hypothesis, we then construct a simple two phase training procedure involving an initial "healing" period followed by a phase of high-quality training. In *two phase* training, we train for 26 billion tokens of FineWeb-Edu followed by the same data as seen in the single phase training, totaling 52 billion tokens. We note it is common to heal models after pruning to regain language modeling performance (Yang et al., 2024; Men et al., 2024).

In Figure 7, we visualize accuracy on GSM8K over training for the 26 billion tokens on the combination of FineWeb-Edu, Nemotron-Pretraining-SFT-v1-General, and Nemotron-Pretraining-SFT-v1-Math data, i.e. the secondary phase after healing for the two phase approach. We see that under single phase training, when we directly train on the mix of all three datasets, the final recurrent model is worse than the non-recurrent model. Next, we observe that during two phase training, the non-recurrent model does not perform much differently than during single phase training. Intuitively, this could be explained by the fact that the initial model is already trained for 3 trillion tokens of web text and as there is no retrofitting process performed on the non-recurrent baseline, there is nothing to explicitly "heal." However, for our depth-recurrent model, two phase training provides an increase of over 9% on GSM8K, demonstrating that the initial 26 billion token healing period is effective in helping the model to regain basic language modeling abilities. Our results demonstrate that **a good data curriculum helps depth-recurrent models maintains and improves language modeling performance while improving on grade school math problems.**

In Table 1, we measure zero-shot accuracy on Arc-Easy (Clark et al., 2018), Arc-Challenge (Clark et al., 2018), Hellaswag (HS) (Zellers et al., 2019), Winogrande (WG) (Sakaguchi et al., 2021), MMLU (Hendrycks et al., 2020), PIQA (Bisk et al., 2020), and OpenbookQA (OBQA) (Mihaylov et al., 2018). We see that the depth-recurrent model achieves high scores across all benchmarks only outperformed by the non-recurrent model on MMLU by less than one standard error. In Table 1, we also see, our models are competitive with the much larger `Huginn-0125` model released by Geiping et al. (2025) achieving an MMLU score over 12% higher and GSM8K performance over 7% higher than their published evaluation results. We include comparisons to the base TinyLlama, as well as evaluations using more test recurrences for our models in Appendix Table 6. Overall, we find **depth-recurrent models can increase accuracy on GSM8K while improving or maintaining performance across a broad range of language modeling benchmarks and simultaneously reducing unique parameter count.**

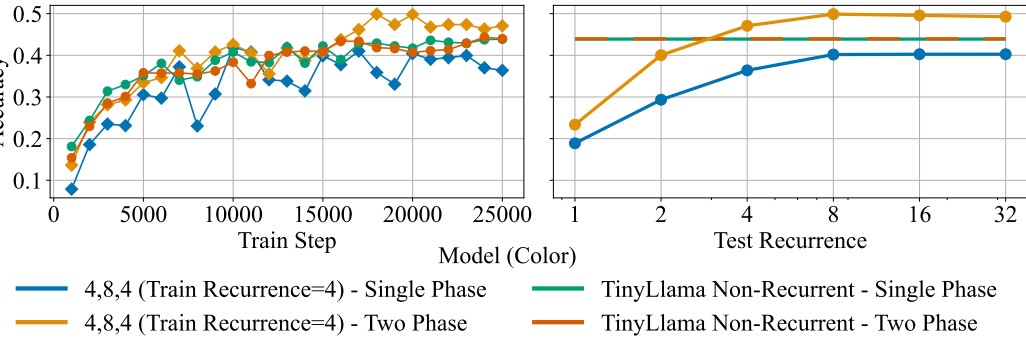

Figure 7: **High quality data and curricula improve recurrent model performance on GSM8K. Left:** We plot accuracy on GSM8K over training for the 26 billion tokens on FineWeb-Edu and Nemotron-SFT data, i.e. after healing for two phase training. We see the training accuracy of the non-recurrent model does not differ significantly between single or two phase training. For the depth-recurrent model, two phase training outperforms single phase by 9% suggesting the healing period helps the model recover language modeling ability after retrofitting. **Right:** Accuracy over multiple recurrences at the end of training. We see the depth-recurrent model with two phase training can use recurrence to extend its accuracy to 49.9% by utilizing more FLOPs during inference. We repeat our GSM8K accuracies in the final column of Table 1 for clarity at test recurrences 1 and 32.

Table 1: **High quality data and curricula improve recurrent model performance across benchmarks.** We see that depth-recurrence achieves better accuracy when using two phase training and confirm that the depth-recurrent models improve as a function of test-time recurrence. On the other hand, for the non-recurrent baseline we see single phase and two phase training perform similarly. Full results in Appendix Table 6, including test recurrences 2, 4, 8 and 16.
** We note our context restricted and without chat template evaluations decrease performance of Huginn-0125, hence we do not reevaluate the model under our conditions and instead state the best accuracies released by Geiping et al. (2025), and do not include them in bolding. We note that this model has over $4\times$ as many parameters as our $(4, 8, 4)$ models.

| Test Recurrence | Arc-E | Arc-C | HS | WG | MMLU | PIQA | OBQA | GSM8K |
|---|---|---|---|---|---|---|---|---|
| 4,8,4 (Train Recurrence=4) - Single Phase | | | | | | | | |
| 1 | 50.0 | 31.6 | 50.8 | 58.0 | 35.7 | 69.3 | 38.8 | 18.9 |
| 32 | 52.7 | 32.7 | 58.2 | **61.1** | 39.4 | 71.4 | 38.6 | 40.3 |
| 4,8,4 (Train Recurrence=4) - Two Phase | | | | | | | | |
| 1 | 52.7 | 31.6 | 51.5 | 56.7 | 36.2 | 71.0 | 39.4 | 23.4 |
| 32 | **65.2** | **37.7** | **60.4** | 60.5 | 44.8 | **73.6** | **40.0** | **49.3** |
| TinyLlama-1.1b-3T Non-Recurrent - Single Phase | | | | | | | | |
| | 61.2 | 35.2 | 58.9 | 60.5 | **45.1** | 71.4 | 39.2 | 43.9 |
| TinyLlama-1.1b-3T Non-Recurrent - Two Phase | | | | | | | | |
| | 62.5 | 36.5 | 60.3 | 59.6 | 44.4 | 72.9 | 39.4 | 44.0 |
| Huginn-0125** (Geiping et al., 2025) | | | | | | | | |
| 1 | 34.9 | 24.1 | 29.3 | 49.4 | 23.6 | 55.3 | 26.8 | 0.0 |
| 32 | 69.9 | 38.2 | 65.2 | 59.4 | 31.4 | 76.2 | 38.8 | 42.08 |

## 5 DISCUSSION

Our work demonstrates that depth-recurrent language models are efficient and highlights how train-time and test-time compute are decoupled for recurrent models. We show that pretrained initializations achieve lower loss than scalable initialization schemes (Figure 2) and that training using a recurrence curriculum maintains performance while vastly improving training speed (Figure 3). Our main experiments demonstrate that retrofitting pretrained Llama-3.2, TinyLlama, and OLMo models with recurrence improves their performance on GSM8K and MATH compared to a static depth baseline (Figures 5 and 6). Finally, we show that including a healing period can help recover basic language modeling functions before training on high quality task specific data (Table 1). Overall, we are able to build recurrent models that beat parameter equivalent baselines and even compete with Huginn-0125 (Geiping et al., 2025) a pre-existing recurrent depth model with $4\times$ the parameters.

**Future Work.** Here we identify several promising avenues for future work. Firstly, while we demonstrate pretrained depth recurrence is effective, combining depth recurrence with reinforcement learning objectives may lead to even larger gains, especially when attempting to imbue recurrent models with native adaptivity that automatically assigns the right amount of compute (recurrence) to a given problem based on how difficult it is. Another unsolved problem, is how to most effectively build depth-recurrent models that can recur deeper at test time to solve harder problems than were seen during training, this is also seen by Geiping et al. (2025). Such built-in stopping criteria would in principle allow models to think deeply on hard problems while solving easy problems quickly, strongly related to early exiting (Elbayad et al., 2019). Figure 1 and Figure 11 present our search process on selecting which layers to keep and which ones to discard, but future work could identify a more optimal method for layer choice during retrofitting, as we find current methods (Men et al., 2024) for removing layers from models to be suboptimal for our recurrent post training goal. Finally, our experiments are at the 1B parameter and 50B token scales, so more experimentation is required to verify that our method generalizes to much larger model and data scales. Scaling laws may also provide insight into the promise of this direction in the long term.

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

## APPENDIX

### ETHICS STATEMENT

We train using open source models and data. Permissively licensed language models have been released on these datasets previously, suggesting the additional risk this work poses is small. A deeper understanding of how to efficiently train models does have many potential positive impacts as it reduces carbon emissions and it lowers the bar of entry into language model training.

### REPRODUCIBILITY STATEMENT

We provide our training code and example Hugging Face compatible modeling files in the supplementary material. Our code is based on that of Geiping et al. (2025) and therefore carries the same Apache 2.0 license. Our example training code points to Huginn-0125 (Geiping et al., 2025) by default, since we cannot upload the model weights trained in this work to openreview due to size limitations. Instead, we provide modeling files with code to retrofit pretrained models to be recurrent models to support reproducibility.

## A  EXTENDED RELATED WORKS

The field of methods for leveraging adaptive test-time computation with architectural modifications (e.g. von Oswald et al., 2025) and additional training methodologies (e.g. Guo et al., 2025) is vast and we refer the reader to Zhu et al. (2025) for a detailed survey.

Recurrent models have been a cornerstone of machine learning for many years (Amari, 1972; Hopfield, 1982; Gers & Schmidhuber, 2000; Sutskever et al., 2008). depth-recurrent architectures can all be viewed as learning the gradient of an energy based model (LeCun & Huang, 2005). Gladstone et al. (2025) show energy based models can be scaled effectively. Recurrent mechanisms are shown to learn generalizable solutions to problems using ResNet (He et al., 2015) based architectures (Schwarzschild et al., 2021; Bansal et al., 2022; Anil et al., 2022; Schwarzschild, 2023; Bear et al., 2024).

Yang et al. (2023); Giannou et al. (2023); Gatmiry et al. (2024) and Fan et al. (2024) study the potential theoretical benefits of recurrence at small scales. Many works study the impact of depth for transformers both theoretically and practically (Levine et al., 2020; Merrill et al., 2022; McLeish et al., 2025; Zuo et al., 2025; Merrill & Sabharwal, 2025; Csordás et al., 2025), it is still an open question how recurrent depth impacts the performance of transformers. Saunshi et al. (2025) demonstrate the power of recurrence by showing chain of thought (Wei et al., 2022) steps can be implicitly simulated in latent space using recurrence. Similar to latent thinking is continuous chain of thought (Hao et al., 2024), a finetuning method to add recurrent behavior to pretrained language models, but training is limited as it requires sequential computations.

Prior work on model surgery has heavily studied converting pretrained transformer language models into linear complexity attention models (Kasai et al., 2021; Zhang et al., 2024a; Mercat et al., 2024; Wang et al., 2024).

## B    ADDITIONAL TECHNICAL DETAILS

**Optimization**    Similarly to Geiping et al. (2025), we train all models with truncated backpropagation (Williams & Peng, 1990; Mikolov et al., 2011), only recording gradients for at most the last 8 uses of the recurrent block. We train in `bfloat16` mixed precision (Zamirai et al., 2020), with Flash Attention (Dao, 2023) and compile the model when training. Notably, to compile the model at scale we observe repeating the prebuilt inductor cache on each individual node removes deadlock errors and improves speed. We train all models on AMD MI300A accelerators (AMD, 2023), using distributed data parallel training. We use a warmup-stable-decay learning rate scheduler (Zhai et al., 2022; Geiping & Goldstein, 2023), adjusting the warmup and decay periods to be appropriate for each experiment. We optimize with the official implementation of Muon[2]. Muon shards the Newton-Schulz calculations between all accelerators and then communicates them, overcoming some of the efficiency degradations compared to Adam. Combined with the fact that the models we are optimizing are smaller language models, we do not observe a degradation in step time when using Muon.

## C    ADDITIONAL EXPERIMENTS

### C.1    RETROFITTING ABLATIONS

In Figure 8, we perform a linear extrapolation of the loss curves shown in Figure 2, seeing the extrapolations intersect at approximately 950 billion tokens. We note this is more than likely an underestimate as there is still curvature in the loss curves. In Figure 9, we continue training the models from Figure 2, cooling the learning rate down over an additional 12 billion tokens. In Figure 10, we vary the `emb_scale` hyperparameter used by Geiping et al. (2025). "Ours" is using the `emb_scale` from the `Huginn-0125` model, where as the line for Geiping et al. (2025) has been adjusted for this specific model shape. We see a negligible difference. In Table 2, we extend Figure 2 with additional test recurrences for other language modelling tasks.

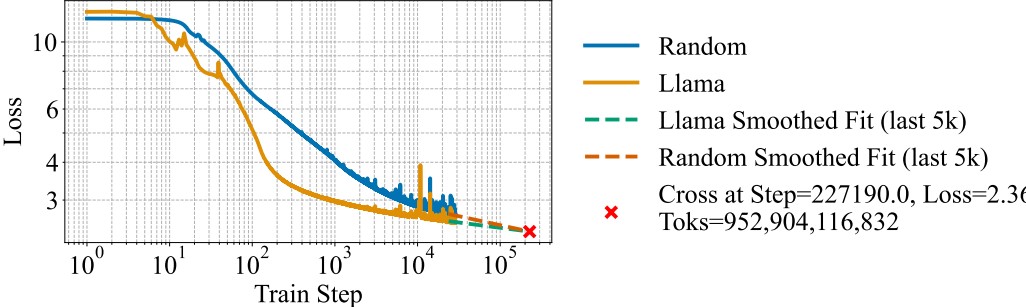

Figure 8: **Training loss for models initialized from Llama-3.2 layers and Randomly.** Here we extend Figure 2 including the linear extrapolations in log-log space. We note this is more than likely an underestimate of the point of intersection as there is still curvature in the loss curves.

---

[2]https://github.com/KellerJordan/Muon

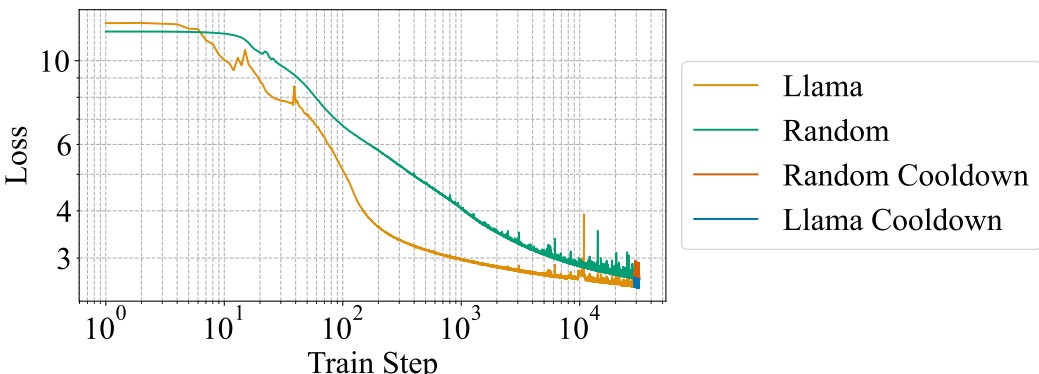

Figure 9: **Training loss for models initialized from Llama-3.2 layers and Randomly.** Here, we extend Figure 2 by including a cooldown for $12b$ additional tokens, taking this to a total of $132b$ tokens.

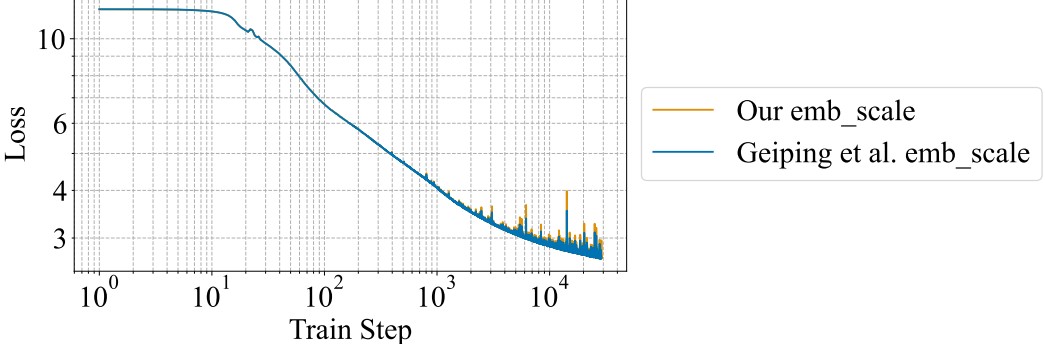

Figure 10: **Training loss for models initialized Randomly with different embedding scales over** 120 **billion tokens.** We follow Geiping et al. (2025) when initializing models with scaled embeddings. However, we also ablate how much the scale impacts by using the same embedding scale from Huginn-0125 in this much smaller model, we find there to be minimal impact.

Table 2: **Initializing from pretrained model weights yields consistent gains across benchmarks.** We evaluate our models trained for 120 billion tokens in a zero shot setting seeing clear advantages to initializing from pretrained weights.

| Test Recurrence | Arc-E | Arc-C | HS | WG | MMLU | PIQA | OBQA |
|---|---|---|---|---|---|---|---|
| Random | | | | | | | |
| | 25 | 25 | 25 | 50 | 25 | 50 | 25 |
| Takase init | | | | | | | |
| 1 | 36.1 | 23.4 | 28.6 | 50.5 | 22.9 | 55.2 | 26.6 |
| 2 | 41.2 | 22.4 | 31.6 | 50.2 | 23.0 | 58.3 | 28.4 |
| 4 | 50.7 | 26.7 | 37.8 | 48.8 | 23.4 | 63.4 | 31.2 |
| 8 | 54.5 | 29.4 | 45.4 | 53.3 | 24.4 | 67.9 | 35.8 |
| 16 | 55.8 | 30.0 | 47.8 | 53.7 | 24.8 | 68.7 | 36.6 |
| 32 | 56.1 | 29.5 | 48.3 | 54.3 | 25.0 | 68.9 | 36.8 |
| Llama init | | | | | | | |
| 1 | 41.6 | 23.8 | 34.8 | 51.3 | 22.9 | 62.5 | 27.2 |
| 2 | 48.4 | 26.6 | 41.2 | 51.4 | 23.2 | 65.9 | 30.6 |
| 4 | 54.5 | 30.8 | 49.4 | 53.2 | 24.0 | 69.7 | 35.4 |
| 8 | 59.2 | 34.0 | 54.9 | 55.6 | **25.4** | 72.3 | 38.4 |
| 16 | 60.2 | **35.1** | 55.4 | 55.7 | 25.3 | **73.1** | 38.4 |
| 32 | **60.4** | 35.0 | **55.6** | **56.1** | 25.3 | 72.9 | **38.6** |

### C.1.1 WHICH LAYERS TO TAKE?

In Figure 11, we compare the layers we found to be optimal to dropping the least impactful layers using the ShortGPT method (Men et al., 2024). We find for training recurrent depth models our selection is better. In Figure 12, we show additional results for models with more varied shapes.

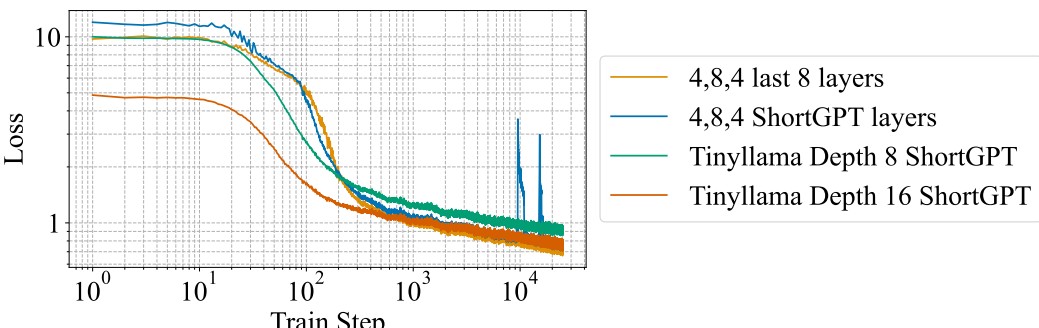

Figure 11: **Comparison to prior methods for decreasing depth.** We use the ShortGPT pruning method proposed by Men et al. (2024) to decrease the depth of the TinyLlama model. We train two non-recurrent models with this pruning method, reducing TinyLlama's depth to 8 and 16. We also train a 4, 8, 4 model using our layer selection (See Table 9 and a model using the layers prescribed by ShortGPT. We train on the nemotron dataset for approximately 25 billion tokens and find our layer selection to be better in terms of loss.

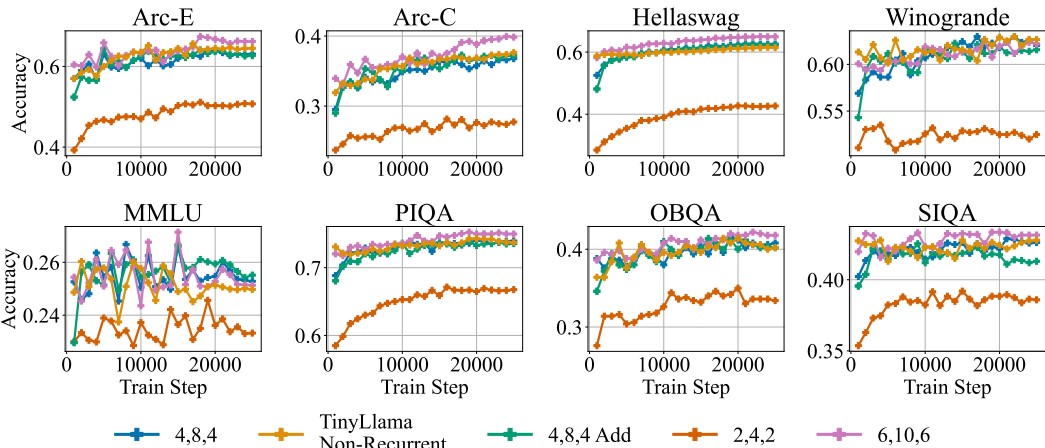

Figure 12: **We ablate different layer selections and architectural choices for TinyLlama.** We show the accuracy on evaluations at 32 recurrences after training on Fineweb-Edu. We ablate $(2, 4, 2)$, $(4, 8, 4)$ and $(6, 10, 6)$ models with $(6, 10, 6)$ keeping all of the layers of the depth 22 TinyLlama model. It is clear to see that keeping all the layers allows for the model to achieve higher accuracy, consistently beating the fixed depth model. Hence, removing some layers does cause a degradation in possible performance. We also ablate swapping the linear adapter used by Geiping et al. (2025) and in our main results for an addition adapter ("Add"). We find that although training loss is higher the evaluation accuracy is approximately the same.

### C.2 SCHEDULING RECURRENCES ABLATIONS

In Figure 13, we visually show the values our curriculum takes, looking like a staircase from 1 to the maximum value over the curriculum period. In Figure 14, we extend Figure 3, showing more curriculum lengths and more test recurrences. In Figure 15, we show the result of scheduling the backpropagation depth over training.

In Figure 16, we show various curricula for models initialized from scratch.

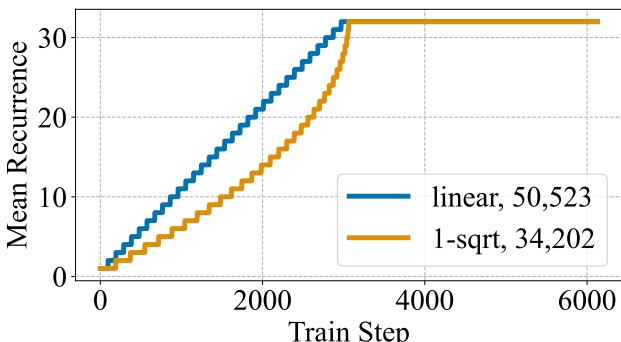

Figure 13: **Visualization of our curriculum over training steps.** We visualize a curriculum with 3125 steps over a training period of 6250 steps with a final mean recurrence of 32. We show both a *linear* and 1-*sqrt* schedules.

$f_{\text{linear}}(\text{tgt\_depth}, \text{current\_step}) = ceil(\text{tgt\_depth} * (\text{current\_step}/\text{num\_warmup\_steps}))$

$f_{\text{1-sqrt}}(\text{tgt\_depth}, \text{current\_step}) = ceil(\text{tgt\_depth}*(1-sqrt(1-\text{current\_step}/\text{num\_warmup\_steps})))$

In the legend we include the number of recurrences used during the curriculum period, seeing the 1-sqrt schedule uses fewer recurrences.

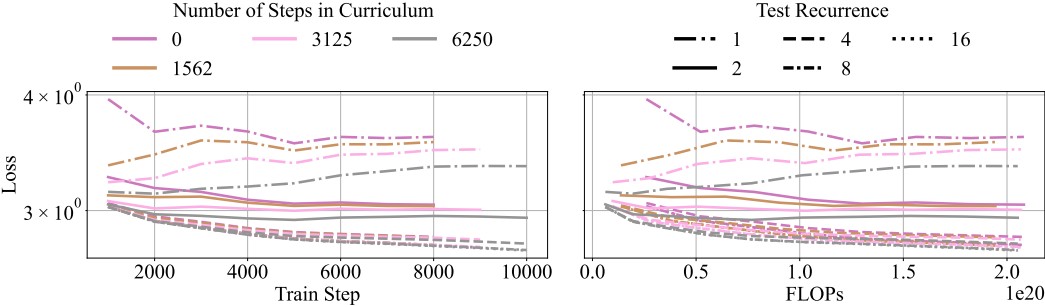

Figure 14: **Scheduling the mean of the depth distribution is efficient in terms of data and compute.** We extend Figure 3, showing more curriculum lengths on the left and more test recurrences on the right. We see the same as in Figure 3, that it is efficient in terms of data (steps) and compute to schedule the mean of the depth distribution.

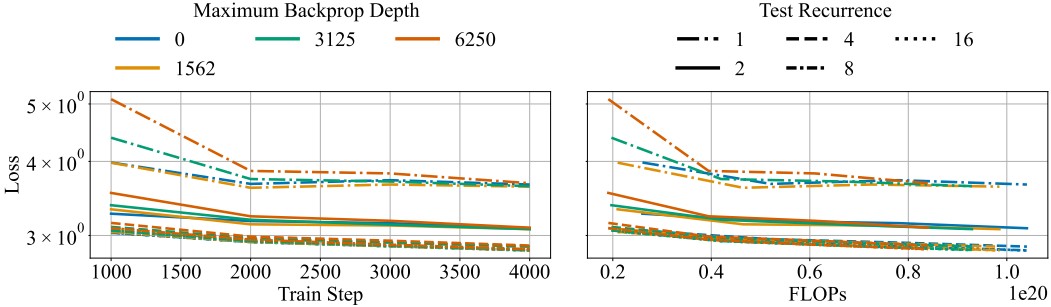

Figure 15: **Validation loss of models with schedules maximum backpropagation depth.** We see that scheduling the maximum backpropagation is efficient in terms of FLOPs spent but does lead to worse models in terms of steps.

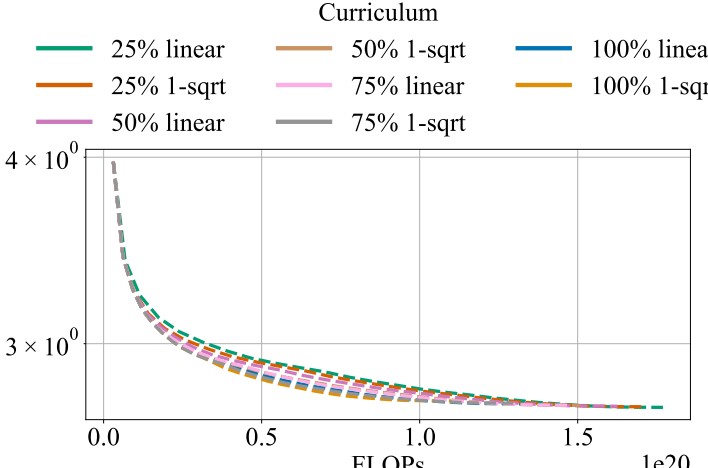

Figure 16: **More aggressive curricula achieve lower loss in terms of FLOPs.** Validation loss of $(2, 4, 2)$ Llama-3.2 depth-recurrent models trained and tested at 4 recurrences, using random initialization (Takase et al., 2023). We see more aggressive (larger) or 1-sqrt over linear (see Figure 13), achieve lower loss faster in terms of training FLOPs.

### C.3    HOW TO RETROFIT RECURRENCE ABLATIONS

#### C.3.1    TINYLLAMA

In Figure 17 we extend Figure 5, showing more train recurrences. In Figure 18, we plot Right of Figures 5 and 17 with an effective parameters x-axis, which can be viewed as proportional to FLOPs required for inference. In Figures 19 and 20 we show the GSM8K accuracy over training step for train recurrences 4, 8, 16 and 32. In Table 3, we show a broad range of evaluations for the models in Figures 5 and 17.

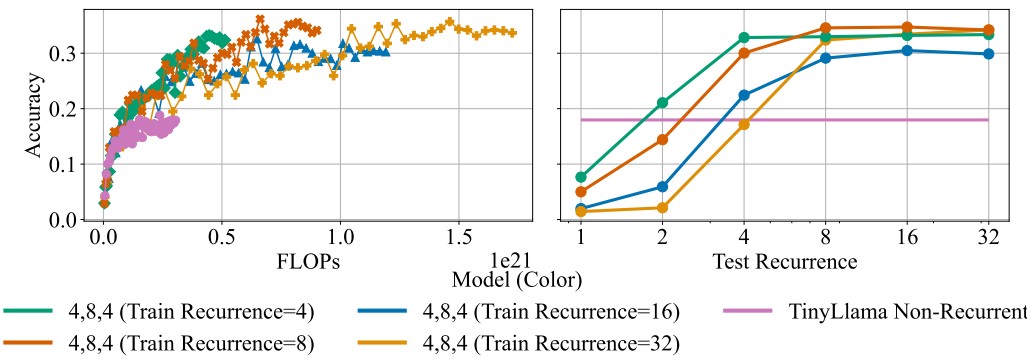

Figure 17: **Recurrence efficiently improves reasoning on GSM8K for TinyLlama.** We train $(4, 8, 4)$ and non-recurrent models for approximately 50 billion tokens of Nemotron-CC-Math-v1 data, extending Figure 5. **Left:** We plot accuracy over the number of FLOPs used during training. We see that recurrent models can efficiently outperform the non-recurrent baseline. **Right:** We plot accuracy over the number of recurrences for inference. We see the recurrent models are competitive with the fixed depth baseline and can outperform it by using more FLOPs.
We plot each individual models accuracy over training and recurrence in full in Figure 19 and Figure 20. Evaluations on the final checkpoint over tasks shown in Table 1 are in Appendix Table 3.

Table 3: **Final step accuracy for models shown in Figure 5 on a broad range of evals.** We also include *TinyLlama-1.1b-3T Hugging Face* which is our evaluations of the TinyLlama-1.1b-3T model downloaded from Hugging Face, i.e. the step 0 accuracy of the non-recurrent TinyLlama model.

| Test Recurrence | Arc-E | Arc-C | HS | WG | MMLU | PIQA | OBQA |
|:---:|:---:|:---:|:---:|:---:|:---:|:---:|:---:|
| 4,8,4 (train rec=4) | | | | | | | |
| 1 | 51.0 | 31.0 | 39.5 | 53.5 | 27.7 | 63.8 | 31.2 |
| 2 | 55.7 | 34.6 | 42.9 | 54.0 | 30.6 | 65.6 | 31.6 |
| 4 | 56.9 | 35.7 | 44.3 | 54.3 | 32.5 | 66.1 | 31.8 |
| 8 | 57.2 | 35.8 | 44.7 | 54.4 | 32.6 | 65.9 | 32.4 |
| 16 | 57.3 | **35.9** | 44.7 | 54.7 | 32.6 | 66.3 | 32.4 |
| 32 | 57.3 | **35.9** | 44.6 | 54.6 | 32.6 | 66.4 | 32.4 |
| 4,8,4 (train rec=8) | | | | | | | |
| 1 | 49.4 | 31.7 | 38.1 | 52.6 | 25.3 | 62.7 | 30.4 |
| 2 | 52.7 | 34.6 | 41.9 | 55.0 | 31.6 | 65.1 | 31.4 |
| 4 | 56.0 | 34.6 | 44.3 | 54.7 | 33.9 | 66.5 | 32.4 |
| 8 | 57.2 | 35.4 | 44.8 | 54.1 | 34.8 | 66.8 | 33.0 |
| 16 | 57.3 | 35.8 | 44.9 | 53.8 | 34.7 | 66.6 | 33.0 |
| 32 | 57.4 | 35.8 | 44.9 | 54.1 | 34.7 | 66.6 | 33.0 |
| 4,8,4 (train rec=16) | | | | | | | |
| 1 | 45.4 | 26.5 | 36.9 | 52.6 | 25.3 | 63.0 | 30.4 |
| 2 | 48.7 | 30.4 | 41.4 | 53.2 | 29.2 | 64.4 | 31.8 |
| 4 | 53.8 | 32.8 | 44.3 | 57.1 | 32.2 | 65.4 | 33.6 |
| 8 | 56.6 | 34.5 | 45.0 | 56.1 | 34.8 | 65.9 | 32.8 |
| 16 | 56.9 | 34.0 | 45.0 | 55.6 | 34.8 | 65.7 | 32.8 |
| 32 | 56.8 | 34.1 | 45.1 | 55.2 | 34.6 | 65.7 | 33.0 |
| 4,8,4 (train rec=32) | | | | | | | |
| 1 | 42.3 | 25.5 | 34.5 | 49.9 | 25.3 | 59.9 | 28.2 |
| 2 | 47.1 | 31.4 | 39.2 | 51.3 | 27.7 | 63.8 | 30.8 |
| 4 | 52.6 | 33.8 | 43.9 | 53.9 | 33.0 | 65.8 | 32.6 |
| 8 | 57.4 | 34.7 | 45.0 | 54.9 | 35.4 | 66.9 | 34.6 |
| 16 | 58.4 | 35.4 | 45.0 | 54.8 | **36.0** | 66.8 | 33.4 |
| 32 | **58.5** | 35.5 | 44.9 | 54.9 | **36.0** | 66.7 | 33.6 |
| TinyLlama-1.1b-3T Non-Recurrent | | | | | | | |
| | 57.5 | 34.9 | 45.3 | 55.8 | 33.4 | 68.8 | 32.8 |
| TinyLlama-1.1b-3T Hugging Face | | | | | | | |
| | 55.7 | 31.0 | **59.1** | **58.9** | 25.4 | **73.0** | **35.0** |

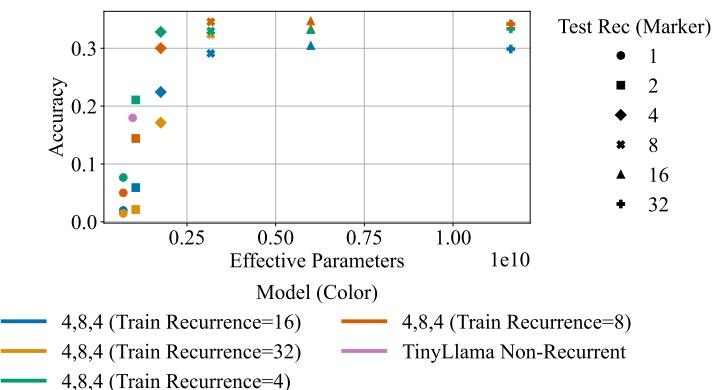

Figure 18: **Recurrent models are competitive in terms of FLOPs.** This is the same data as in Right of Figures 5 and 17 but replotted with an effective parameters x-axis, which can be viewed as proportional to FLOPs required for inference.

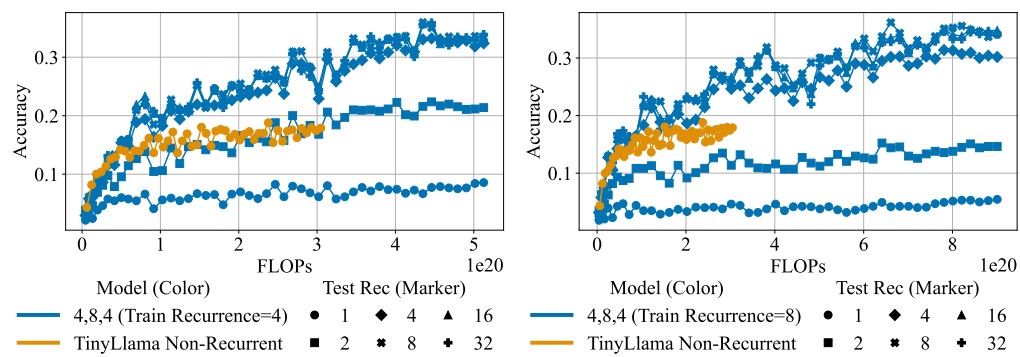

Figure 19: **Recurrence efficiently improves reasoning.Left**: GSM8K accuracy over training step for train recurrence equal to 4 model. **Right**: GSM8K accuracy over training step for train recurrence equal to 8 model.

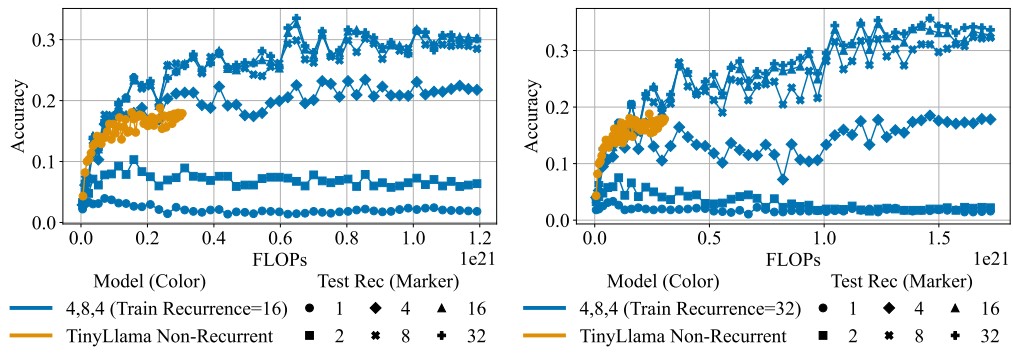

Figure 20: **Recurrence efficiently improves reasoning.Left**: GSM8K accuracy over training step for train recurrence equal to 16 model. **Right**: GSM8K accuracy over training step for train recurrence equal to 32 model.

### C.3.2 LLAMA

In Figure 21 we extend Figure 6, showing more train recurrences. In Figure 22, we plot Right of Figures 6 and 21 with an effective parameters x-axis, this can be viewed as proportional to FLOPs

required for inference. In Figures 23 and 24 we show the GSM8K accuracy over training step for train recurrences 4, 8, 16 and 32. In Table 4, we show a broad range of evaluations for the models in Figures 6 and 21.

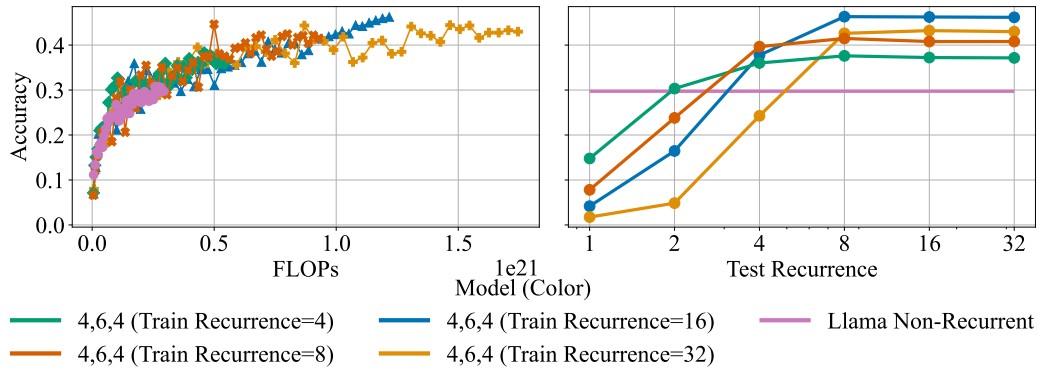

Figure 21: **Recurrence efficiently improves reasoning on GSM8K for Llama-3.2.** We train $(4, 6, 4)$ and non-recurrent models for approximately 50 billion tokens of Nemotron-CC-Math-v1 data, extending Figure 5. **Left:** We plot accuracy over the number of FLOPs used during training. We see that recurrent models can efficiently outperform the non-recurrent baseline. **Right:** We plot accuracy over the number of recurrences for inference. We see the recurrent models are competitive with the fixed depth baseline and can outperform it by using more FLOPs.

We plot each individual models accuracy over training and recurrence in full in Figure 23 and Figure 24. Evaluations on the final checkpoint over tasks shown in Table 1 are in Appendix Table 4.

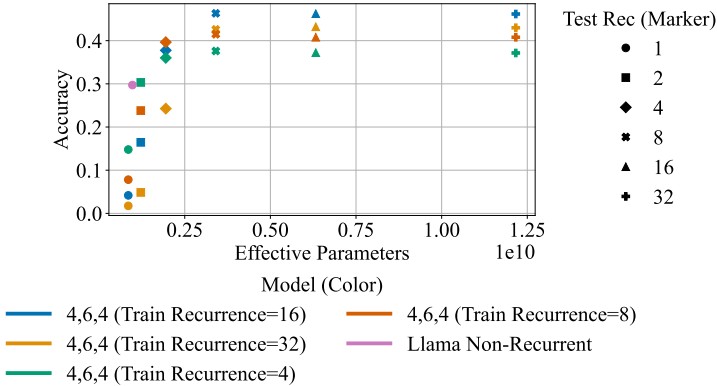

Figure 22: **Recurrent models are competitive in terms of FLOPs.** This is the same data as in Right of Figures 6 and 21 but replotted with an effective parameters x-axis, which can be viewed as proportional to FLOPs required for inference.

Table 4: **Final step accuracy for models shown in Figure 6 on a broad range of evals.** We also include *Llama-3.2-1B Hugging Face* which is our evaluations of the Llama-3.2-1B model downloaded from Hugging Face, i.e. the step 0 accuracy of the non-recurrent Llama-3.2 model.

| Test Recurrence | Arc-E | Arc-C | HS | WG | MMLU | PIQA | OBQA |
|---|---|---|---|---|---|---|---|
| 4,6,4 (train rec=4) | | | | | | | |
| 1 | 58.5 | 34.4 | 42.3 | 54.6 | 33.6 | 67.1 | 31.6 |
| 2 | 60.8 | 37.9 | 44.7 | 55.2 | 35.9 | 67.5 | 33.6 |
| 4 | 61.3 | 38.7 | 45.3 | 57.0 | 36.8 | 67.7 | 34.0 |
| 8 | 61.7 | 38.7 | 45.4 | 56.8 | 36.9 | 67.6 | 33.8 |
| 16 | 61.7 | 38.7 | 45.4 | 56.7 | 36.8 | 67.8 | 34.0 |
| 32 | 61.7 | **38.7** | 45.4 | 56.8 | 36.9 | 67.8 | 34.0 |
| 4,6,4 (train rec=8) | | | | | | | |
| 1 | 54.8 | 33.2 | 41.4 | 51.8 | 32.0 | 66.4 | 33.2 |
| 2 | 58.8 | 36.9 | 44.7 | 52.6 | 35.5 | 67.2 | 35.6 |
| 4 | 60.8 | 37.5 | 46.0 | 53.7 | 38.1 | 68.1 | 35.6 |
| 8 | 60.9 | 38.2 | 46.2 | 54.0 | 38.4 | 67.8 | 35.8 |
| 16 | 61.1 | 38.4 | 46.2 | 53.8 | 38.3 | 68.1 | 36.0 |
| 32 | 61.2 | 38.3 | 46.2 | 53.7 | 38.3 | 68.1 | 36.0 |
| 4,6,4 (train rec=16) | | | | | | | |
| 1 | 52.8 | 30.4 | 39.8 | 51.5 | 28.3 | 66.2 | 31.4 |
| 2 | 58.3 | 35.8 | 43.8 | 53.8 | 33.2 | 67.2 | 33.6 |
| 4 | 60.1 | 37.6 | 46.1 | 54.9 | 36.9 | 68.4 | 32.8 |
| 8 | 60.2 | 38.1 | 46.5 | 54.9 | 37.5 | 68.8 | 32.6 |
| 16 | 60.1 | 38.0 | 46.5 | 54.8 | 37.6 | 68.9 | 32.2 |
| 32 | 60.1 | 38.1 | 46.6 | 54.7 | 37.7 | 68.9 | 32.2 |
| 4,6,4 (train rec=32) | | | | | | | |
| 1 | 48.1 | 26.4 | 36.8 | 49.3 | 26.5 | 62.6 | 28.0 |
| 2 | 56.4 | 33.8 | 41.7 | 52.8 | 30.2 | 65.7 | 32.6 |
| 4 | 60.8 | 36.3 | 45.7 | 54.9 | 36.5 | 68.1 | 31.8 |
| 8 | 61.2 | 36.6 | 46.4 | 55.6 | 38.9 | 68.1 | 32.4 |
| 16 | 61.5 | 36.4 | 46.4 | 55.5 | 38.8 | 68.3 | 33.0 |
| 32 | 61.7 | 36.4 | 46.4 | 55.3 | **38.9** | 68.2 | 33.0 |
| Llama-3.2-1B Non-Recurrent | | | | | | | |
| - | **62.6** | 38.2 | 45.8 | 57.1 | 38.7 | 68.4 | 33.4 |
| Llama-3.2-1B Hugging Face | | | | | | | |
| - | 61.7 | 36.9 | **64.2** | **60.9** | 38.6 | **74.9** | **37.2** |

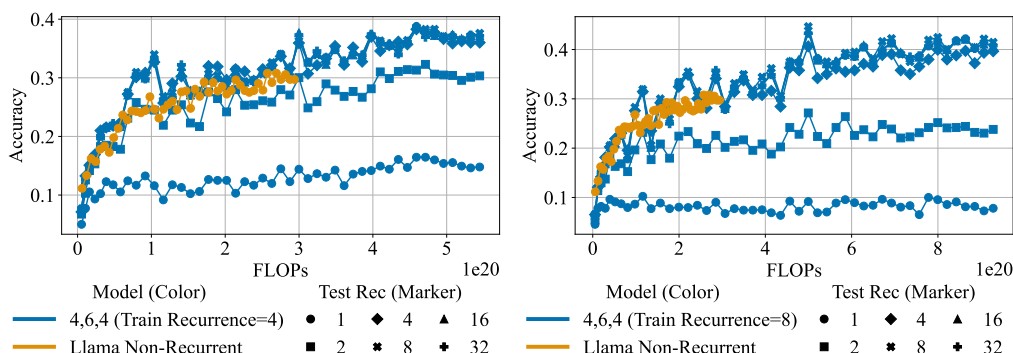

Figure 23: **Recurrence efficiently improves reasoning.Left**: GSM8K accuracy over training step for train recurrence equal to 4 model. **Right**: GSM8K accuracy over training step for train recurrence equal to 8 model.

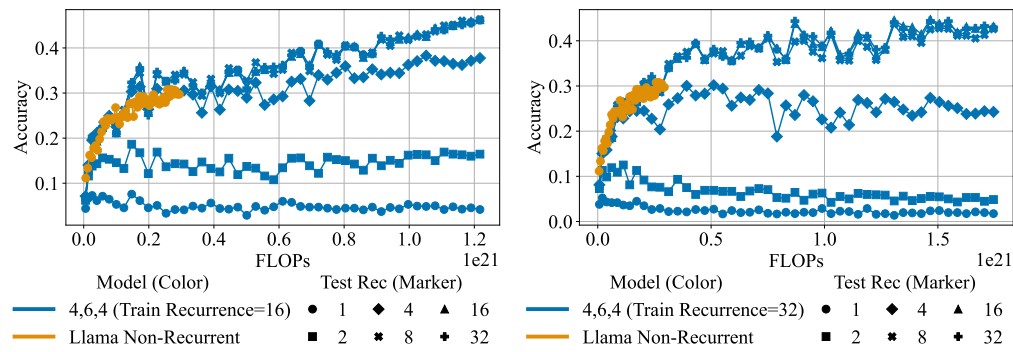

Figure 24: **Recurrence efficiently improves reasoning.Left**: GSM8K accuracy over training step for train recurrence equal to 16 model. **Right**: GSM8K accuracy over training step for train recurrence equal to 32 model.

### C.3.3   OLMO

For OLMo, we construct $(4, 6, 4)$ configurations, removing 2 layers (layers 4 and 5 with 0 indexing) from the pretrained model. This leaves approximately 900 million remaining parameters in the recurrent model, which equates to $87.5\%$ of the pretrained models parameters. We note the OLMo models use QK-norm and a post-normalization scheme unlike the Llama models which use a pre-normalization scheme and do not use a QK-norm. We train OLMo models on approximately 50 billion tokens of `Nemotron-CC-Math-v1-4plus` (Mahabadi et al., 2025) data, with a 1-sqrt curriculum (see Figure 13) for the first $75\%$ of training and constant mean recurrences thereafter.

In Figure 25 we show evaluation results for OLMo on GSM8k. In Figure 26, we plot Right of Figure 25 with an effective parameters x-axis, which can be viewed as proportional to FLOPs required for inference. In Figures 27 and 28 we show the GSM8K accuracy over training step for train recurrences $4, 8, 16$ and $32$.

In Table 5, we show a broad range of evaluations for the models in Figure 25.

Table 5: **Final step accuracy for models shown in Figure 25 on a broad range of evaluations.** We also include *OLMo-2-0425-1B-step1907359 Hugging Face* which is our evaluations of the OLMo-2-0425-1B-step1907359 model downloaded from Hugging Face, i.e. the step 0 accuracy of the non-recurrent OLMo model.

| Test Recurrence | Arc-E | Arc-C | HS | WG | MMLU | PIQA | OBQA | GSM8K |
|---|---|---|---|---|---|---|---|---|
| 4,6,4 (Train Recurrence=4) | | | | | | | | |
| 1 | 61.6 | 36.3 | 46.4 | 56.8 | 36.4 | 68.4 | 33.6 | 24.3 |
| 2 | 63.8 | 37.7 | 48.3 | 58.2 | 37.9 | 69.5 | 35.6 | 37.4 |
| 4 | 63.8 | 37.4 | 48.8 | 57.7 | 38.3 | 69.6 | 36.2 | 38.7 |
| 8 | 63.7 | 37.4 | 49.0 | 57.2 | 38.2 | 69.9 | 35.8 | 39.5 |
| 16 | 63.6 | 37.3 | 49.0 | 57.1 | 38.1 | 70.0 | 35.8 | 39.4 |
| 32 | 63.6 | 37.3 | 49.0 | 57.2 | 38.1 | 70.0 | 35.8 | 40.6 |
| 4,6,4 (Train Recurrence=8) | | | | | | | | |
| 1 | 60.9 | 37.0 | 45.9 | 55.8 | 35.2 | 69.2 | 32.0 | 20.5 |
| 2 | 64.0 | 39.1 | 48.4 | 58.4 | 37.5 | 69.7 | 34.4 | 37.6 |
| 4 | 64.8 | 39.6 | 49.2 | 59.9 | 39.0 | 70.3 | 34.4 | 43.1 |
| 8 | 65.0 | 39.7 | 49.4 | 59.3 | 39.2 | 70.6 | 34.4 | 44.4 |
| 16 | 65.0 | 39.4 | 49.5 | 59.4 | 39.1 | 70.5 | 34.2 | 43.6 |
| 32 | 65.0 | 39.4 | 49.5 | 59.4 | 39.1 | 70.5 | 34.2 | 44.6 |
| 4,6,4 (Train Recurrence=16) | | | | | | | | |
| 1 | 57.6 | 35.6 | 45.3 | 56.0 | 33.7 | 68.1 | 36.8 | 18.0 |
| 2 | 62.3 | 38.7 | 48.7 | 59.3 | 37.1 | 67.8 | 34.0 | 36.2 |
| 4 | 64.1 | 40.2 | 49.8 | 58.6 | 39.4 | 69.4 | 34.2 | 46.6 |
| 8 | 65.2 | 39.5 | 49.8 | 58.6 | 39.9 | 69.8 | 34.6 | 48.4 |
| 16 | 65.2 | 39.8 | 49.8 | 58.0 | 39.8 | 70.0 | 34.8 | 48.7 |
| 32 | 65.2 | 39.8 | 49.8 | 58.0 | 39.8 | 70.0 | 34.8 | 48.3 |
| 4,6,4 (Train Recurrence=32) | | | | | | | | |
| 1 | 56.9 | 33.2 | 44.4 | 54.1 | 30.9 | 66.8 | 33.4 | 10.4 |
| 2 | 61.7 | 37.8 | 47.9 | 55.5 | 36.5 | 68.1 | 34.2 | 30.6 |
| 4 | 65.2 | 38.9 | 49.4 | 59.2 | 39.3 | 68.8 | 33.2 | 44.6 |
| 8 | 66.0 | 39.8 | 49.6 | 58.2 | 40.4 | 69.6 | 34.4 | 48.6 |
| 16 | 66.1 | 40.2 | 49.8 | 57.8 | **40.5** | 69.9 | 34.4 | 49.7 |
| 32 | 66.0 | 40.2 | 49.8 | 57.5 | **40.5** | 70.0 | 34.4 | **51.6** |
| Olmo-2 Non-Recurrent | | | | | | | | |
| | 65.2 | **40.8** | 50.7 | 60.0 | 40.0 | 70.0 | 35.4 | 35.3 |
| OLMo-2-0425-1B-step1907359 Hugging Face | | | | | | | | |
| | **67.6** | 39.2 | **67.0** | 65.3 | 24.6 | **76.3** | **39.2** | 3.6 |

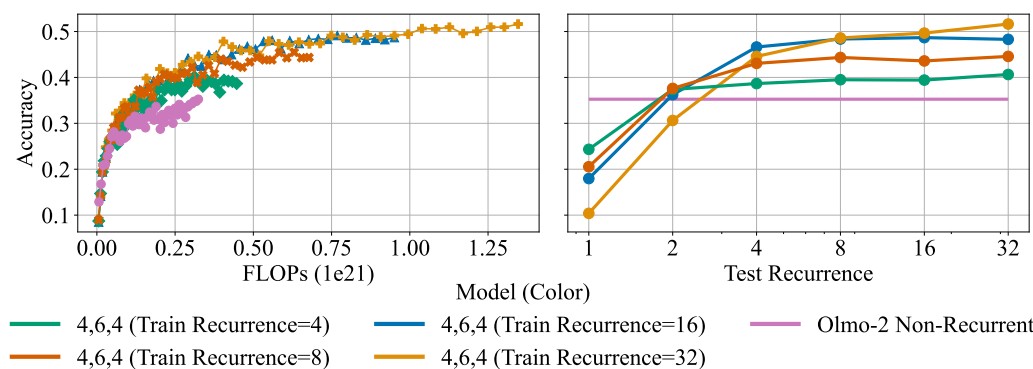

Figure 25: **Recurrence efficiently improves reasoning on GSM8K for OLMo.** We train $(4, 8, 4)$ and non-recurrent models for approximately 50 billion tokens of Nemotron-CC-Math-v1 data. **Left:** We plot accuracy over the number of FLOPs used during training. We see that recurrent models can efficiently outperform the non-recurrent baseline. **Right:** We plot accuracy over the number of recurrences for inference. We see the recurrent models are competitive with the fixed depth baseline and can outperform it by using more FLOPs.

We plot each individual models accuracy over training and recurrence in full in Figure 27 and Figure 28. Evaluations on the final checkpoint over tasks shown in Table 1 are in Appendix Table 5.

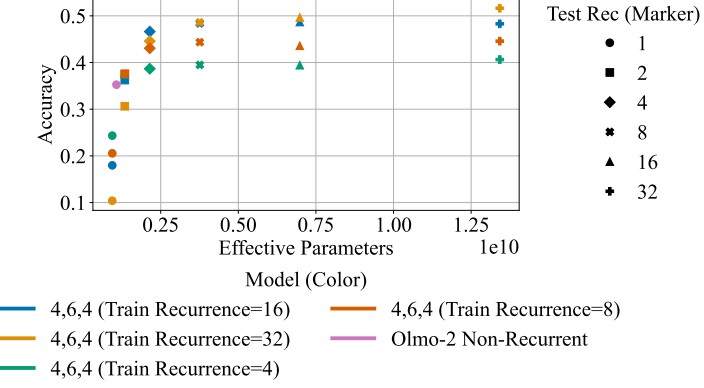

Figure 26: **Recurrent models are competitive in terms of inference FLOPs for GSM8K.** This is the same data as in 25 but replotted with an effective parameters x-axis, which can be viewed as proportional to FLOPs required for inference.

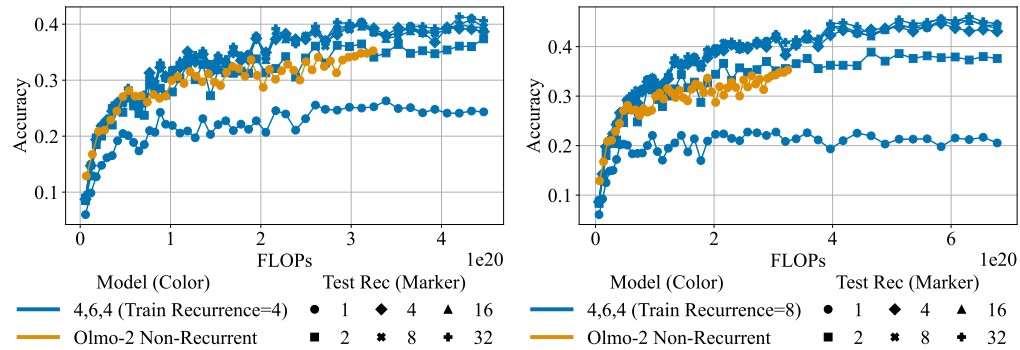

Figure 27: **Recurrence efficiently improves reasoning. Left**: GSM8K accuracy over training step for train recurrence equal to 4 model. **Right**: GSM8K accuracy over training step for train recurrence equal to 8 model.

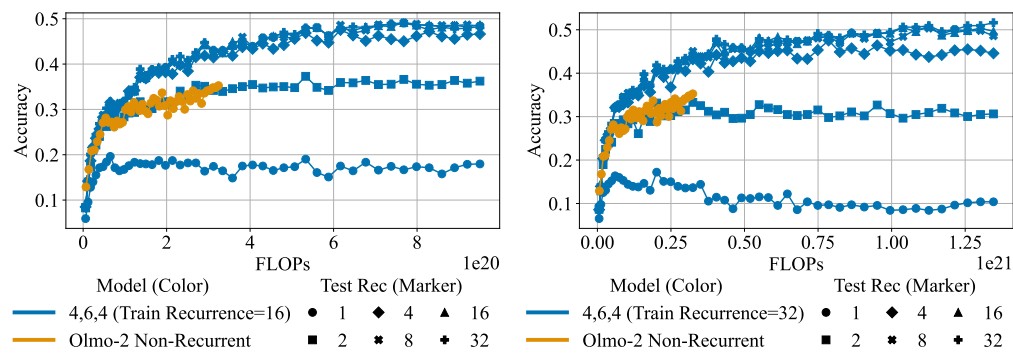

Figure 28: **Recurrence efficiently improves reasoning. Left**: GSM8K accuracy over training step for train recurrence equal to 16 model. **Right**: GSM8K accuracy over training step for train recurrence equal to 32 model.

### C.3.4 MAJORITY VOTING

In this section, we compare to static depth baselines using majority voting (Dietterich, 2000; Trad & Chehab, 2024). We use temperature 0.2 for all evaluations in this section to allow for stochasticity in completions for majority voting.

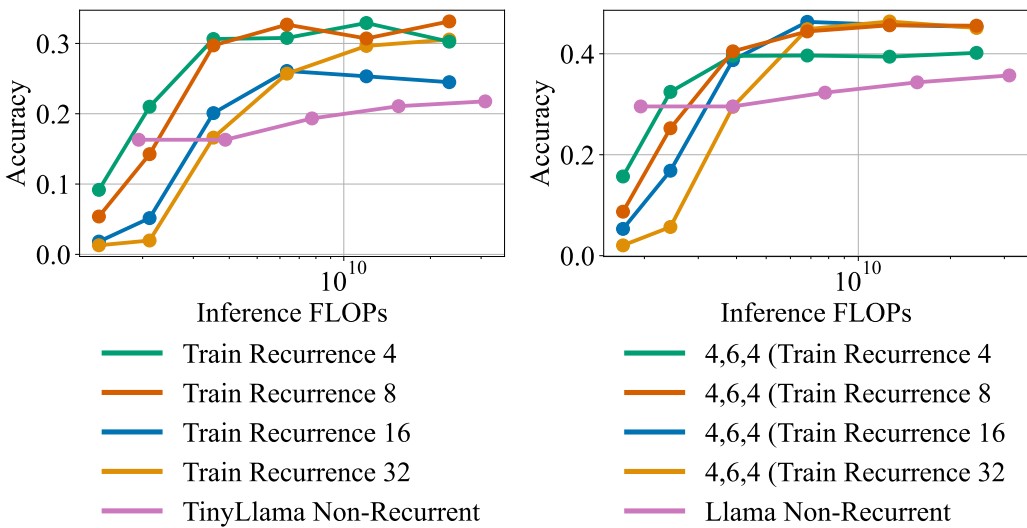

Figure 29: **Recurrent models achieve higher GSM8K accuracy than static-depth models using majority voting.** Static-depth models are evaluated with majority voting over 1,2,4,8, and 16 completions, while recurrent models are evaluated without majority voting but with test-time recurrence scaling. We observe that depth-recurrent models use test-time compute more efficiently, achieving higher accuracy with fewer inference FLOPs per token. Left: TinyLlama. Right: Llama-3.2.

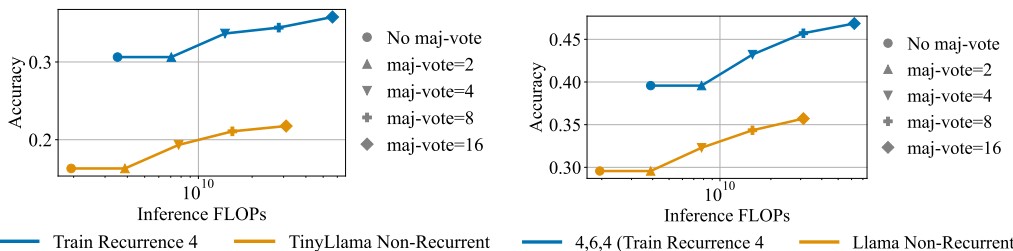

Figure 30: **Recurrent and static-depth models achieve higher GSM8K accuracy with majority voting.** Accuracy of recurrent and static-depth models on GSM8K when using majority voting over 1,2,4,8, and 16 completions. All models show improved accuracy with more votes, corresponding to increased inference FLOPs per token. The recurrent model is evaluated with test recurrence = 4. **Left**: TinyLlama. **Right**: Llama-3.2.

### C.4 DATA MIXTURES

In Table 6, we extend Table 1, including more test recurrences. We also include the Huginn-0125 evaluations conducted and published by Geiping et al. (2025) for comparison.

Table 6: **High quality data and curricula improve recurrent model performance across benchmarks.** We see that the depth-recurrent models increases in accuracy over recurrence and achieves better accuracy when using two phrase training. For the non-recurrent baseline we see single phase training slightly outperforms two phrase training. This table extends Table 1.
** We note our context restricted and without chat template evaluations would more than likely decrease performance of Huginn-0125, hence we do not reevaluate the model under our conditions and instead state the accuracies released by Geiping et al. (2025). We note that this model has over $4\times$ as many parameters as our $(4, 8, 4)$ models.

| Test Recurrence | Arc-E | Arc-C | HS | WG | MMLU | PIQA | OBQA | GSM8K |
|---|---|---|---|---|---|---|---|---|
| | | | | Random | | | | |
| | 25 | 25 | 25 | 50 | 25 | 50 | 25 | 0 |
| | | 4,8,4 (Train Recurrence=4) - Single Phase | | | | | | |
| 1 | 50.0 | 31.6 | 50.8 | 58.0 | 35.7 | 69.3 | 38.8 | 18.9 |
| 2 | 52.3 | 31.9 | 55.8 | 60.5 | 39.2 | 70.9 | 38.8 | 29.3 |
| 4 | 53.3 | 32.8 | 57.7 | 60.9 | 39.6 | 71.3 | 39.0 | 36.4 |
| 8 | 52.5 | 32.4 | 58.1 | 60.8 | 39.5 | 71.2 | 38.6 | 40.2 |
| 16 | 52.7 | 32.8 | 58.2 | 61.0 | 39.4 | 71.2 | 38.6 | 40.3 |
| 32 | 52.7 | 32.7 | 58.2 | 61.1 | 39.4 | 71.4 | 38.6 | 40.3 |
| | | 4,8,4 (Train Recurrence=4) - Two Phase | | | | | | |
| 1 | 52.7 | 31.6 | 51.5 | 56.7 | 36.2 | 71.0 | 39.4 | 23.4 |
| 2 | 59.3 | 34.8 | 57.3 | 58.6 | 41.3 | 71.3 | 41.0 | 40.0 |
| 4 | 63.8 | 36.9 | 60.0 | 58.7 | 44.3 | 73.5 | 40.6 | 47.1 |
| 8 | 65.2 | 37.4 | 60.3 | 59.9 | 44.7 | 73.7 | 40.0 | 49.9 |
| 16 | 65.2 | 37.7 | 60.4 | 60.2 | 44.8 | 73.6 | 40.0 | 49.6 |
| 32 | 65.2 | 37.7 | 60.4 | 60.5 | 44.8 | 73.6 | 40.0 | 49.3 |
| | | TinyLlama-1.1b-3T Non-Recurrent - Single Phase | | | | | | |
| | 61.2 | 35.2 | 58.9 | 60.5 | 45.1 | 71.4 | 39.2 | 43.9 |
| | | TinyLlama-1.1b-3T Non-Recurrent - Two Phase | | | | | | |
| | 62.5 | 36.5 | 60.3 | 59.6 | 44.4 | 72.9 | 39.4 | 44.0 |
| | | TinyLlama-1.1b-3T (Zhang et al., 2024b) | | | | | | |
| | 55.7 | 31.0 | 59.1 | 58.9 | 25.4 | 73.0 | 35.0 | 1.7 |
| | | Huginn-0125** – 3.5b parameters (Geiping et al., 2025) | | | | | | |
| 1 | 34.9 | 24.1 | 29.3 | 49.4 | 23.6 | 55.3 | 26.8 | 0.0 |
| 32 | 69.9 | 38.2 | 65.2 | 59.4 | 31.4 | 76.2 | 38.8 | 42.08 |

# D  HYPERPARAMETERS

We use a learning rate of $5e^{-5}$ for AdamW and $0.001$ for Muon with weight decay of $1e^{-4}$. We clip all gradients at 1. We use a microbatch size of 8, global batch size of 1024 using 8 nodes of 4 AMD MI300A GPUs (AMD, 2023) by default. For the experiments shown in Section 4.1 and Appendix C.1 we use a global batch size of 4096 on 64 nodes. For experiments shown in Section 4.2 and Appendix C.2 we use a global batch size of 512 on 1 node. When using EllisAdam, we use the same values as Geiping et al. (2025) for all hyper parameters which are not learning rate or weight decay.

# E  PARAMETER COUNTS

In Table 7, we give exact parameter counts for non recurrent models. In Table 8, we give exact parameter counts for recurrent models. In Table 9, we detail the layers we take from the pretrained models to form our depth-recurrent models.

Table 7: Exact parameter counts for non-recurrent models.

| Model Name | Embeddings | Body |
|---|---|---|
| TinyLlama-1.1B-intermediate-step-1431k-3T | $131,072,000$ | $968,976,384$ |
| Llama-3.2-1B (untied) | $525,336,576$ | $973,146,112$ |
| OLMo-2-0425-1B | $411,041,792$ | $1,073,874,944$ |

Table 8: Exact parameter counts for depth-recurrent models.

| Model Name | Embeddings | Body | Prelude | Rec Block | Coda |
|---|---|---|---|---|---|
| TinyLlama $(2, 4, 2)$ | $131,072,000$ | $486,572,032$ | $121,643,008$ | $243,286,016$ | $121,643,008$ |
| TinyLlama $4, 8, 4$ | $131,072,000$ | $704,708,608$ | $176,177,152$ | $352,354,304$ | $176,177,152$ |
| TinyLlama $(6, 10, 6)$ | $131,072,000$ | $968,974,336$ | $264,265,728$ | $440,442,880$ | $264,265,728$ |
| Llama-3.2 $(4, 6, 4)$ | $525,336,576$ | $851,501,056$ | $243,286,016$ | $364,929,024$ | $243,286,016$ |
| OLMo $(4, 6, 4)$ | $411,041,792$ | $939,638,784$ | $268,468,224$ | $402,702,336$ | $268,468,224$ |

Table 9: Layers taken from original non-recurrent models to form depth-recurrent models.

| Model Name | Body | Prelude | Rec Block |
|---|---|---|---|
| TinyLlama $(2, 4, 2)$ | $[0, 1]$ | $[16, 17, 18, 19]$ | $[20, 21]$ |
| TinyLlama $4, 8, 4$ | $[0, 1, 2, 3]$ | $[10, 11, 12, 13, 14, 15, 16, 17]$ | $[18, 19, 20, 21]$ |
| TinyLlama $(6, 10, 6)$ | $[0, 1, 2, 3, 4, 5]$ | $[6, 7, 8, 9, 10, 11, 12, 13, 14, 15]$ | $[16, 17, 18, 19, 20, 21]$ |
| Llama-3.2 $(4, 6, 4)$ | $[0, 1, 2, 3]$ | $6, 7, 8, 9, 10, 11$ | $[12, 13, 14, 15]$ |
| OLMo $(4, 6, 4)$ | $[0, 1, 2, 3]$ | $[6, 7, 8, 9, 10, 11]$ | $[12, 13, 14, 15]$ |

