# OpenReview forum: "Teaching Pretrained Language Models to Think Deeper with Retrofitted Recurrence"
_ICLR.cc/2026/Conference — Submitted to ICLR 2026_

### Official Review · Reviewer_3XYK · 2025-10-21

**Soundness:** 3
**Presentation:** 2
**Contribution:** 2
**Rating:** 4
**Confidence:** 4

**Summary:**

The paper investigates recurrent language models. The main contribution is
- It showed copying weights from pretrained models helps in optimization
- It showed a curriculum over recurrent depths can improve training speed
- It showed TinyLlama and Llama can be converted into recurrent models that have better GSM8K performance compared to vanilla models and can recover basic language modeling performance by a "healing" period.

**Strengths:**

The paper showed recurrent models can have better param efficiency, achieve better results on GSM8K and recover basic language abilities. The experiments is supportive and the claim is clear.

**Weaknesses:**

W1: The biggest one is I do not know what is the paper/your method's position on test-time scaling. I am not saying that recurrent models must surpass other methods like RL-based methods, you can even prove that it is sub-optimal. The paper just lacks this part.

W2: The second weakness is that I do not get enough insights from the paper. The fact that copying weights from pretrained LLMs can have advantage compared to training from scratch is also happening in diffusion language models (Dream). Your second contribution is also in the same case, I think the method is already existing for a long time, or it is an engineering trick...

I think the interesting point/insights in my opinion may be why the layers should be chosen in that way for example. The paper now looks like a technical report to me.

The weaknesses below are minor.

W3: How do you compare recurrent models initialized from pretrained-weights to LLMs fairly, for example you should also take flops that used to train LLMs into account? Also, I think you actually need a larger dataset to pretrain the LLM or get a comparable results if training recurrent models from scratch, which is also a hidden cost.

**Questions:**

Q1: I am curious that what on earth is the difference between: CoT, continuous CoT (Coconut, you do not discrete the vector to tokens), recurrent models (if viewing intermediate recurrent results as CoT) and looped transformers. Are some concepts the same?

Q2: What is the support for the flops calculation. I do not find any experiments or theory that support it.

Q3: It is weird that you trained on fewer recurrence but have better extrapolation (TinyLlama) and I am not satisfied by your explanation. Also, I am curious about the behavior of intermediate choice of #recurrence.

---

> ### Author Response · Authors · 2025-11-17
> **Author Rebuttal Part 1/3**
>
> Thank you for your feedback. We appreciate that you highlight our clear claims and supportive evidence.
>
> In addition to addressing your feedback, during the rebuttal period we have also added OLMo-2-1b models to our results and evaluations on GSM8K, finding that efficient performance improvements can also be achieved on models which are not part of the Llama-3.2 and TinyLlama families (see Appendix Figure 25).
> We believe these additional results further strengthen our findings and demonstrate their robustness across model families.
>
> > W1: I do not know what is the paper's position on test-time scaling.
>
> In principle, depth recurrence and other test time compute schemes may be complementary. One such test time compute scheme, which has been used to great effect in the literature [1,2,3], is majority voting.
> During the rebuttal period, we ran additional evaluations.
> We use majority voting to expend test time compute for the feedforward models and compare to recurrent models on an inference FLOPs per token axis in Figure 29 for TinyLlama and Llama-3.2 models. We see that even when given slightly more inference FLOPs per token, the feedforward (static-depth) models still achieve a much lower accuracy than the recurrent models.
>
> We note that we can also use majority voting for our recurrent models, now visualized in Figure 30. We see similar performance increases for majority voting as the static-depth baselines for both TinyLlama and Llama-3.2. Since the literature contains numerous examples of test time compute approaches, future work might explore the compatibility between these approaches and depth-recurrence. Such an extensive study is beyond the scope of our work. We have included our new evaluations using majority voting in our updated draft.
>
> > W2: The fact that copying weights from pretrained LLMs can have advantage compared to training from scratch is also happening in diffusion language models (Dream). Your second contribution is also in the same case, I think the method is already existing for a long time, or it is an engineering trick...
>
> We agree that reusing the parameters of a pretrained model to initialize other architectures has been shown to work in various domains.  However, the exact details of how this conversion is performed strongly impact downstream performance, and such details vary across applications.  For example, Dream modified the attention mask of a causal transformer and made large performance improvements over LLaDA.  In our work, we devised a scheme to retrofit static-depth language models into recurrent ones, and we show that the precise details are essential.  We further develop a post training recipe for depth recurrent language models, including a data mix and curriculum, and we show that our recipe noticeably outperforms baselines.  We also want to emphasize that engineering and implementation details are crucial to the field of machine learning, and such efforts have been key components of progress.
> For example, DR. GRPO [4] makes slight modifications to existing RL training loops that yield empirical benefits and have been widely adopted, and the Pre-LayerNorm [5] formulation shifts the normalization placement in Transformers, an architectural change now used in most modern language models.
>
>
> [1] Lewkowycz, A., Andreassen, A., Dohan, D., Dyer, E., Michalewski, H., Ramasesh, V., Slone, A., Anil, C., Schlag, I., Gutman-Solo, T. and Wu, Y., 2022. Solving quantitative reasoning problems with language models. Advances in neural information processing systems, 35, pp.3843-3857.
>
> [2] Brown, B., Juravsky, J., Ehrlich, R., Clark, R., Le, Q.V., Ré, C. and Mirhoseini, A., 2024. Large language monkeys: Scaling inference compute with repeated sampling. arXiv preprint arXiv:2407.21787.
>
> [3] OpenAI (2024) Learning to reason with LLMs. Available at: https://openai.com/index/learning-to-reason-with-llms/
>
> [4] Liu, Z., Chen, C., Li, W., Qi, P., Pang, T., Du, C., Lee, W.S. and Lin, M., 2025. Understanding r1-zero-like training: A critical perspective. arXiv preprint arXiv:2503.20783.
>
> [5] Xiong, R., Yang, Y., He, D., Zheng, K., Zheng, S., Xing, C., Zhang, H., Lan, Y., Wang, L. and Liu, T., 2020, November. On layer normalization in the transformer architecture. In International conference on machine learning (pp. 10524-10533). PMLR.

---

> ### Author Response · Authors · 2025-11-17
> **Author Rebuttal Part 2/3**
>
> > I think the interesting insights may be why the layers should be chosen in that way.
>
> We agree this is a very interesting point and have moved this into the main body as Figure 1 to emphasize its importance. In Appendix C.1.1, we find current methodologies for removing layers from models fall short when the objective is to further post train the model to be depth recurrent and highlight this as future work in the discussion section.
> Counterintuitively, recent papers [6] suggest later layers in models are less useful, yet we find them to be highly important under our recurrent post training objective.
>
> > W3: How do you compare recurrent models initialized from pretrained-weights to LLMs fairly?
>
> We note that pretraining a feedforward model and then adding depth-recurrence corresponds to a hyperparameter setting of our curriculum.
> That is, we start with a small number of iterations, potentially as few as 1, and increase the iteration count during training.
> So this point could be rephrased as, what is the right curriculum to train a recurrent model? To further address your question, during the rebuttal period, we train new models, like the ones in Figure 2, from scratch on 25 billion tokens of FineWeb-Edu using a variety of curriculum periods and using a more aggressive 1-sqrt schedule (see updated Figure 13). The 1-sqrt schedule spends longer at lower recurrence counts during the curriculum period than the linear schedule. We present our new findings in Figure 16, visualizing the validation loss at 4 recurrences for our models. We see that the most aggressive 100% 1-sqrt schedule, meaning the model is trained feedforward for the longest period of time, is the most FLOPs efficient in terms of validation loss. Although a larger scale experiment, perhaps over trillions of tokens, is required to fully validate this hypothesis, this experiment suggests that training a feedforward model and then increasing depth via recurrence may be FLOPs optimal.
>
> In our work, one of our goals is to produce a high performing recurrent model. Given our compute limitations, it is useful for us to begin with a strong open weights model.
>
> We do not currently have any evidence to support or reject your claim that recurrent depth models require larger datasets for pretraining. However, our paper does show that this is not the case for post training and using the same dataset always leads to higher GSM8K performance for the recurrent depth model.
>
> > Q1: I am curious that what is the difference between: CoT, continuous CoT (Coconut, you do not discrete the vector to tokens), recurrent models (if viewing intermediate recurrent results as CoT) and looped transformers. Are some concepts the same?
>
> We begin with brief descriptions of each method below, please refer to the associated links for a full description of each method:
> - CoT: reasoning is performed in discrete token space, and each iteration involves sampling a new token which in turn grows the KV-Cache. (see https://arxiv.org/pdf/2201.11903).
> - Continuous CoT (CCoT): the reasoning is done in a latent space, but each iteration still involves producing a new token which in turn grows the KV-Cache (see https://arxiv.org/pdf/2412.06769).
> - Recurrent Models, Looped Transformers, and Universal Transformers: In the literature, these refer to approximately the same thing, where the reasoning is also done in latent space for these models but where the reasoning only increases the computation depth and does not generate new tokens, hence not growing the KV-Cache (see https://arxiv.org/pdf/2502.05171).
>
> Depth-recurrent language models have several benefits. Firstly, they do not require the whole model to be repeated for each step in the thought process, unlike CoT and CCoT. Second, because depth-recurrent methods do not grow the KV-Cache, they are significantly more memory efficient.
>
> > Q2: What is the support for the flops calculation?
>
> We use the same arithmetic used by Kaplan et al. (https://arxiv.org/pdf/2001.08361), and have updated our draft to make this clearer. On the forward pass 2 floating point operations are computed per parameter and on the backward pass 4 floating point operations are computed per parameter. This gives rise to the approximation FLOPs=6ND. As we do 6 computations per parameter (N), a number of tokens times (D).
> We use the same method, counting 2 computations if a parameter is used without gradients (i.e. no backward is computed) and 6 if a parameter is used with gradients. Giving rise to $FLOPs=(6(\text{params used with grad}) + 2(\text{params without grad}))D$
>
> [6] Sun, W., Song, X., Li, P., Yin, L., Zheng, Y. and Liu, S., 2025. The curse of depth in large language models. arXiv preprint arXiv:2502.05795.

---

> ### Author Response · Authors · 2025-11-17
> **Author Rebuttal Part 3/3**
>
> > Q3: It is weird that you trained on fewer recurrence but have better extrapolation (TinyLlama).
>
> Due to compute constraints, we choose recurrences based on powers of $2$, training from $2^2$ through $2^5$, and testing from $2^0$ through $2^5$. For TinyLlama, the Train Recurrence 16 model underperforms all models trained with different train recurrences, however still outperforms the static depth baseline considerably. Please see Figure 17 for the full comparison. We run all experiments in identical conditions with exactly the same data ordering and curriculum period, hence our best hypothesis is that this is due to the different model shapes and pretraining biases, as for Llama-3.2 (Figure 21) and OLMo (added during rebuttal period please see global comment, Figure 25) increasing training recurrence directly leads to improved performance.
> We agree that the non-monotonicity of the trends we observe is an interesting observation, and potential non-monotonicity should be factored into design choices of future work.
>
>
> In response to your feedback, we have made significant efforts to improve our draft, and we have also included extensive new experiments which we think have improved the quality of our work. We would greatly appreciate it if you would consider raising your score accordingly. Do you have any other questions we can address?

---

> > ### Author Response · Authors · 2025-11-24
> >
> > Thanks again for your feedback.  We made a strong effort to address all your points, including new experiments and paper edits, and we would greatly appreciate it if you would consider increasing your score accordingly.  Do you have any other questions?

---

> > > ### Comment · Reviewer_3XYK · 2025-11-26
> > > **Official Comment by the Reviewer**
> > >
> > > I thank the author for the detailed explanations. My concerns are all addressed, therefore I would raise the score accordingly.

---

> > > > ### Author Response · Authors · 2025-11-28
> > > >
> > > > We would like to personally thank you for engaging in the discussion process, increasing your score from 4 to 6 in light of our hard work to improve our draft, including additional experiments and evaluations during the rebuttal period.

---

### Official Review · Reviewer_fYfk · 2025-10-24

**Soundness:** 2
**Presentation:** 3
**Contribution:** 2
**Rating:** 4
**Confidence:** 3

**Summary:**

This paper shows that introducing latent recurrence into pretrained language models can improve the reasoning performance. The main contribution is a set of techniques that translate a feed-forward transformer architecture (from the Llama family) to the latent recurrent model by Geiping et al. (2025). The techniques include weight transfer from pretrained feed-forward transformer, recurrence scheduling during training, optimizer selection, and language capability through “healing” (i.e., train on natural language modeling dataset). Empirically, the paper shows that initializing the recurrent model with pretrained feed-forward weights yields better accuracy compared to random initialization. Moreover, the resulting recurrent architecture is also more performant in math compared to an equally post-trained feed-forward model.

**Strengths:**

This is clearly an empirical paper, and the authors did a reasonable job in describing and conducting the experiments. While not particularly strong on the methodological side, the main insight that reuse of pretrained feed-forward weights for latent recurrent networks is practically useful.

**Weaknesses:**

1. The paper shows that it is beneficial to initialize the weights in a latent recurrent model with pretrained feed-forward weights. However, it is not clear if this approach is overall compute optimal, i.e., FLOPS(pretrain feed-forward)+FLOPS(post-train recurrent) > FLOPS(only pretrain recurrent (maybe for longer)). Hence, we’re still missing a clear compute-optimal recipe for training latent recurrent models.
2. The reasoning results show the feed-forward performances without test-time scaling. It would be beneficial to consider test-time scaling also for these architectures.
3. The comparison to the baseline (only pretrain recurrent) from Geiping et al. (2025) should be included in the main Table 1, and not hidden in the appendix.

**Questions:**

I would appreciate it if the rebuttal could address the individual weaknesses.

---

> ### Author Response · Authors · 2025-11-17
> **Author Rebuttal**
>
> Thank you for your valuable feedback. We address each of your points below.
>
> > The paper shows that it is beneficial to initialize the weights in a latent recurrent model with pretrained feed-forward weights. However, it is not clear if this approach is overall compute optimal.
>
> Thank you for this interesting question.
>
> We note that pretraining a feedforward model and then adding depth-recurrence corresponds to a hyperparameter setting of our curriculum.
> That is, we start with a small number of iterations, potentially as few as 1, and increase the iteration count during training.
> So this point could be rephrased as, what is the right curriculum to train a recurrent model? To further address your question, during the rebuttal period, we train new models, like the ones in Figure 2, from scratch on 25 billion tokens of FineWeb-Edu using a variety of curriculum periods and using a more aggressive 1-sqrt schedule (see updated Figure 13). The 1-sqrt schedule spends longer at lower recurrence counts during the curriculum period than the linear schedule. We present our new findings in Figure 16, visualizing the validation loss at 4 recurrences for our models. We see that the most aggressive 100% 1-sqrt schedule, meaning the model is trained feedforward for the longest period of time, is the most FLOPs efficient in terms of validation loss. Although a larger scale experiment, perhaps over trillions of tokens, is required to fully validate this hypothesis, this experiment suggests that training a feedforward model and then increasing depth via recurrence may be FLOPs optimal.
>
> In our work, one of our goals is to produce a high performing recurrent model. Given our compute limitations, it is useful for us to begin with a strong open weights model.
>
> > The reasoning results show the feed-forward performances without test-time scaling. It would be beneficial to consider test-time scaling also for these architectures.
>
> We thank the reviewer for offering this insight. During the rebuttal period, we ran additional evaluations. We use majority voting to expend test time compute for the feedforward models and compare to recurrent models on an inference FLOPs per token axis in Figure 29 for TinyLlama and Llama-3.2 models. We see that even when given slightly more inference FLOPs per token, the feedforward (static-depth) models still achieve a much lower accuracy than the recurrent models.
>
> We note that we can also use majority voting for our recurrent models, now visualized in Figure 30. We see similar performance increases for majority voting as the static-depth baselines for both TinyLlama and Llama-3.2.
>
> > The comparison to the baseline (only pretrain recurrent) from Geiping et al. (2025) should be included in the main Table 1.
>
> We agree that it is easier to view all results in one table and move this up into the main body.
> We also highlight that although the model from Geiping et al. has more than 4x the number of parameters our models have, we perform competitively, even achieving higher accuracy in some cases.
>
>
> In addition to addressing your feedback, during the rebuttal period we have also added OLMo-2-1b models to our results and evaluations on GSM8K, finding that efficient performance improvements can also be achieved on models which are not part of the Llama-3.2 and TinyLlama families (see Appendix Figure 25)
> We believe these additional results further strengthen our findings and demonstrate their robustness across model families.
>
> We believe that your feedback has significantly improved our updated draft, which now includes additional experiments and evaluations. We kindly ask that you consider updating your score accordingly.
> Do you have any other questions or concerns we can address?

---

> > ### Author Response · Authors · 2025-11-24
> >
> > Thanks again for your feedback.  We made a strong effort to address all your points, including new training and evaluation experiments, and we would greatly appreciate it if you would consider increasing your score accordingly.  Do you have any other questions?

---

> > > ### Comment · Reviewer_fYfk · 2025-11-24
> > >
> > > I thank the authors for the comprehensive rebuttal and the new experiments.
> > >
> > > The analysis of the "1-sqrt schedule" (Figure 16) effectively addresses my concern regarding compute optimality. Additionally, the test-time scaling comparison using majority voting (Figure 29) ensures a fair comparison against feed-forward baselines.
> > >
> > > I strongly suggest including these new results in the main body of the final paper. I am raising my score.

---

> > > > ### Author Response · Authors · 2025-11-25
> > > >
> > > > Thank you for acknowledging our rebuttal, including our new training and evaluation experiments. Unfortunately, we are space constrained in the main body, but have added pointers to the new figures in the main body following your positive feedback.
> > > >
> > > > Thank you for verbally confirming that you will increase your score. We kindly ask that you please edit the score in your original review at your earliest convenience to confirm this.

---

> > > > > ### Author Response · Authors · 2025-11-28
> > > > >
> > > > > We would like to personally thank you for engaging in the discussion process, increasing your score from 4 to 6 in light of our strong effort to improve our draft, including new experiments and evaluations during the rebuttal period.

---

### Official Review · Reviewer_sNb3 · 2025-10-29

**Soundness:** 3
**Presentation:** 1
**Contribution:** 3
**Rating:** 4
**Confidence:** 4

**Summary:**

This paper investigates how to leverage pretrained LLMs when developing recurrent-depth language models. To do so, the authors use the recurrent architecture from Geiping et al. (2025) and take transformer blocks from a pretrained (fixed-depth) Llama to initalize its transformer blocks, rather than random initialization. In order to make this work, the paper presents a two-phase training regime, where the language model capabilities are "recovered" first before math reasoning fine-tuning. At the start of this training, recurrence depth is progressively scaled up. Results on the target task, GSM8K, show that this recurrent-depth language model outperforms random initialization but more importantly, the original language model from which the transformer blocks were taken. In addition, performance on other tasks is maintained or improved.

**Strengths:**

1. clear motivation (expensive training of depth-recurrent models) and a practical idea (leveraging heavily trained Llama models)
1. intuitive method for re-purposing transformer blocks from pretrained fixed-depth LLMs.
1.  effective training regime strategy that notably incorporates a recurrence-scheduling curriculum, adapted from recent works.
1. significant amount of ablations (architectural configuration, layer selection, initialization, optimizer, training phases, data mixtures, etc.) that provide justification for the different components of the methodology
1. the presented end-to-end method both (1) improves upon the fixed-depth model from which the transformer blocks are taken and (2) proves more efficient/better than random initialization.

**Weaknesses:**

Presentation/Paper Organization
1.  The abstract is very insufficient; while concise, it lacks important details that help explain the paper.
1.  The terminology used creates quite a bit of confusion. Notably, the terms "surgery", initialize", "convert", and "retrofit" seem to be used to describe overlapping concepts. For example, retrofit is used to describe the method altogether, but also specifically the retraining part. It took me a long while to understand what was going on because of this.
1. In general, I would settle a single-framing: either this method can be framed as an efficient way to initialize recurrent-depth language models or as a way to convert fixed-depth language models to recurrent-depth ones. It seems as though the authors try to frame this method both ways concurrently.
1. The organization of Section 3 & 4 can be greatly improved. Section 3 vs. 4 should either be split architecture & initialization vs. training or description of experiments/ablations vs. results.
1. the abstract, intro, and discussion emphasize GSM8K, why? The non-GSM8K results (Table 1) are arguably more impressive. I argue the benchmark evaluations should be presented very differently, and perhaps in their own section.
1. Discussion section is poorly organized and doesn't sufficiently bring together findings/contributions of paper and contextualize them in current landscape

In general, the presentation & framing required numerous reads to fully understand.

*Suggestions/Other:*
1. L51-58 these two sentences are not very clear.
1. Related Works should mention some early-exit and speculative decoding literature.
1. L137 I think the empirical experiments of the layer selection should be emphasized more, as if it feels like a critical piece. I would bring up from the appendix the empirical results & provide some intuition (grounded in literature perhaps?)
1. Figure 6 presents single-phase and two-phase side by side with respect to train step, but this is a bit weird since the two-phase would have been trained for 26 billion tokens prior to 0.

**Questions:**

1. L47 this seems like a very arbitrary checkpoint? what is the explanation?
1. Even when training on SFT data, it is treated as unsupervised data for CPT, correct?

---

> ### Author Response · Authors · 2025-11-17
> **Author Rebuttal Part 1/2**
>
> Thank you for your positive feedback on our results and our intuitive and effective methodology.
>
> In addition to addressing your feedback, during the rebuttal period, we have also added OLMo-2-1b models to our results and evaluations on GSM8K, finding that efficient performance improvements can also be achieved on models which are not part of the Llama-3.2 and TinyLlama families (see Appendix Figure 25).
> We believe these additional results further strengthen our findings and demonstrate their robustness across model families.
>
> Moreover, based on feedback from other reviewers, we have added additional experiments exploring how to FLOPs-efficiently train depth-recurrent models from scratch (see Figure 16). We find aggressive curricula, where the model is trains for smaller recurrences for larger periods of training, are most FLOPs efficient.
> Moreover, we explore test time compute for our static-depth models via majority voting (see Figure 29). We see that even when given more inference FLOPs per token, our depth-recurrent models achieve higher GSM8K accuracy than the static-depth models, and and recurrence is broadly a more efficient way to improve performance than majority voting.  Below, we address each of your points.
>
> > The abstract ... lacks important details.
>
> Prompted by your feedback, we have now updated the abstract in our latest draft, emphasizing data mixtures and our models' ability to scale via recurrence at test time.
>
> > The terms "surgery", "initialize", "convert", and "retrofit" seem to be used to describe overlapping concepts.
>
> Thanks for pointing out this inconsistency. We now use the term "initialize" to describe the weights of the model at step 0 of post training, and "retrofit" to describe the process of training the looped models. Any uses of "surgery" or "convert" now only refer to prior work, mirroring the terminology from those prior papers.
>
> > I would settle a single-framing: either this method can be framed as an efficient way to initialize recurrent-depth language models or as a way to convert fixed-depth language models to recurrent-depth ones.
>
> > L51-58 these two sentences are not very clear.
>
> Thank you for pointing out this confusion.
> We have now updated our draft to clarify our precise goals.
> Our central goal is to train highly performing depth-recurrent models in the most compute efficient way possible.
> There are two success metrics we identify:
> 1. We want our model, initialized from pretrained weights, to outperform a model trained from random initialization.
> 2. We want the performance of our recurrent model to be higher than that of the static-depth model we use for initialization.
>
> Ensuring we succeed on both metrics requires two baselines:
> 1. A depth-recurrent model trained from random initialization (Section 4.1).
> 2. The static-depth model, with additional post training for fair comparison (Sections 4.3 and 4.4).
>
> We have now updated lines 51-58 in our draft based on your feedback.
>
> > Section 3 vs. 4 should either be split architecture & initialization vs. training or description of experiments/ablations vs. results.
>
> We have now updated Sections 3 and 4 to closely align with the "architecture & initialization vs. training" split you suggested.
> The rephrasing of lines 51-58, introduction and discussion in line with your wider feedback also makes this clearer. We have also added or improved summary paragraphs at the beginning of each section to emphasize the path through the paper.
>
> > The abstract, intro, and discussion emphasize GSM8K, why? The non-GSM8K results (Table 1) are arguably more impressive.
>
> Thank you for your acknowledgement of our impressive results. The new abstract in response to weakness 1, our updated introduction, and discussion now emphasize the non-GSM8K results as well.
>
> > Discussion section.
>
> The discussion section now highlights future research directions, and we have improved our brief summary in line with the feedback provided in this review. Further, we have added citations to our proposals for future work to be more contextualized with the current state of depth-recurrent language modeling.
>
> > Related Works should mention some early-exit and speculative decoding literature.
>
> We have now updated the related works highlighting early-exiting and speculative decoding.
>
> > I think the empirical experiments of the layer selection should be emphasized more.
>
> We agree and have now moved the empirical experiments for layer selection figure up from the appendix to now be Figure 1.
> In Figure 11, we highlight that current literature on removal of layers falls short when attempting to pick optimal layers for recurrent model post training. Instead we do an empirical search over layers in Figure 1. In our discussion we emphasize this as a point for future work to resolve, and note our empirical search already provides results that are strong compared to fixed-depth baselines.

---

> ### Author Response · Authors · 2025-11-17
> **Author Rebuttal Part 2/2**
>
> > L47 this seems like a very arbitrary checkpoint?
>
> TinyLlama releases base model checkpoints every 500B tokens with this naming format, we take the last one (3T tokens), and give the full name to be clear.
>
> > Even when training on SFT data, it is treated as unsupervised data for CPT, correct?
>
> Yes, we use base model checkpoints and Continued PreTraining (CPT).
>
> We believe your feedback has significantly improved our updated draft. We have conducted extensive experiments and made numerous changes to our draft based on your feedback, and we kindly ask that you consider increasing your score accordingly. Do you have any other questions we can address?

---

> > ### Author Response · Authors · 2025-11-24
> >
> > Thanks again for your feedback.  We made a strong effort to address all your points, including substantial edits to our draft, and we would greatly appreciate it if you would consider increasing your score accordingly.  Do you have any other questions?

---

> > > ### Comment · Reviewer_sNb3 · 2025-11-25
> > > **Response to Author Rebuttal**
> > >
> > > Thank you for addressing so many of my weaknesses. I believe, at least for me, the presentation has greatly improved, notably the ease of following as well as the motivation/narrative. The flow of Sections 3 & 4 works much better, I believe. I am raising my score.
> > >
> > > Further feedback (minor):
> > > - I think the conclusion (or discussion) could better wrap together the takeaways from the paper, all the while discussing future work.

---

> > > > ### Author Response · Authors · 2025-11-25
> > > >
> > > > Thank you for acknowledging the hard work we put in to action your feedback, we are very happy you believe the paper is greatly improved. We have updated the first paragraph of the discussion, in line with your additional feedback.

---

> > > > > ### Author Response · Authors · 2025-11-28
> > > > >
> > > > > We would like to personally thank you for engaging in the discussion process, increasing your score from 4 to 8 in light of our hard work to improve our draft.

---

### Official Review · Reviewer_pTpx · 2025-11-04

**Soundness:** 3
**Presentation:** 3
**Contribution:** 3
**Rating:** 8
**Confidence:** 3

**Summary:**

The paper studies more efficient training of depth-recurrent models (Geipeng et al. 2025). Depth-recurrent models are based on the intuition of test time compute but instead of the model 'thinking' in discrete tokens it 'thinks' in continuous space.

The authors study how to use a pre-initialized non-recurrent model to initialize training for a depth recurrent model showing:

- that it gives significant computational advantages compared to starting from scratch.

-using a curriculum to increase the recurrence depth over training is beneficial

-Finding that Muon is a more effective than AdamW in the authors' use case.

-A two stage training pipeline (where model first trains on FineWeb-Edu before the general mixture), which the authors find benefits the depth recurrent model training (probably due to the 'healing' required after the model surgery when initializing)

Overall I found the paper well written and practically useful.

**Strengths:**

Nice practical study on an important problem given the excitement around depth recurrence and test-time compute in the community.

A solid set of experiments and ablations are provided and I think this paper will be a useful reference to practitioners in the field.

**Weaknesses:**

-I found the terminology, 'Tiny Llama' and 'Llama' to be confusing. I think it would be clearer if 'Llama' also had a prefix to make it clear there are two different models.

**Questions:**

See above.

---

> ### Author Response · Authors · 2025-11-17
> **Author Rebuttal**
>
> Thank you for your valuable feedback. We appreciate that you highlight the excitement in the community surrounding depth recurrence and how useful our paper will be in light of this excitement.
>
> > I think it would be clearer if 'Llama' also had a prefix.
>
> Prompted by your feedback, we now use Llama-3.2 to refer to the Llama-3.2-1b model.
>
> In addition to your feedback, during the rebuttal period we have also added OLMo-2-1b models to our results and evaluations on GSM8K, finding that efficient performance improvements can also be achieved on models which are not part of the Llama-3.2 and TinyLlama families (see Appendix Figure 25)
> We believe these additional results further strengthen our findings and demonstrate their robustness across model families.
>
> Moreover, based on feedback from other reviewers we have added additional experiments exploring how to FLOPs efficiently train depth-recurrent models from scratch (see Figure 16). We find aggressive curricula, where the model is trains for smaller recurrences for larger periods of training, are most FLOPs efficient.
> Moreover, we explore test time compute for our static-depth models via majority voting (see Figure 29). We see that even when given more inference FLOPs per token, our depth-recurrent models achieve higher GSM8K accuracy than the static-depth models.
>
> We believe that your feedback has improved our updated draft, and we ask that you consider increasing your score accordingly.
> Do you have any other questions we can address?

---

> > ### Author Response · Authors · 2025-11-24
> >
> > Thanks again for your feedback, we made a strong effort to address all your points. Do you have any other questions?

---

### Author Response · Authors · 2025-11-17
**Author Rebuttal**

We thank all reviewers for their valuable feedback. We have now addressed all feedback by updating our draft, adding significant new experiments, and expanding evaluations.

We have worked to add OLMo-2-1b models to our results and evaluations on GSM8K, finding that efficient performance improvements can also be achieved on models which are not part of the Llama-3.2 and TinyLlama families (see Appendix Figure 25).
We believe these additional results further strengthen our findings and demonstrate their robustness across model families.

During the rebuttal period, we have added additional experiments exploring how to FLOPs-efficiently train depth-recurrent models from scratch (see Figure 16). We find aggressive curricula, where the model is trains for smaller recurrences for larger periods of training, are most FLOPs efficient.
Moreover, we explore test time compute for our static-depth models via majority voting (see Figure 29). We see that even when given more inference FLOPs per token, our depth-recurrent models achieve higher GSM8K accuracy than the static-depth models, and recurrence is broadly a more efficient way to improve performance than majority voting.

---

### Author Response · Authors · 2025-11-28
**Final Author Remarks**

We thank the reviewers, originally assigned AC, and newly assigned AC for all their efforts during this review process.
We especially appreciate the reviewers' engagement during the rebuttal period. Even though the updates to reviews will no longer be visible, we genuinely believe that the discussion helped us improve the paper.

We again highlight some of our new work from the rebuttal, which addressed all reviewer concerns: expanding our experiments from Llama-3.2-1b and TinyLlama-1b to OLMo-2-1b, seeing strong gains using recurrence; expending test time compute for the static depth baselines using majority voting, seeing recurrence is more effective than majority voting for increasing accuracy with test time compute; and new experiments exploring FLOPs-efficiency when training depth-recurrent models from scratch, finding our novel curriculum method is also impactful in this regime.
Moreover, we worked to improve the readability of our draft, with reviewer `sNb3` finding it greatly improved.

---

### Meta-Review · Area_Chair_hCzx · 2026-01-13

**Summary:**

This paper proposes a "retrofitting" technique to initialize depth-recurrent language models using weights from pretrained static-depth models (specifically Llama-3.2 and TinyLlama). The authors introduce a curriculum learning approach that scales recurrence depth during training and demonstrate performance gains on reasoning tasks like GSM8K compared to standard post-training of static models.

While the paper is interesting, my primary concern of the paper is on the lack of novelty. The core concept of depth-recurrent models and "thinking" in continuous space has been explored in prior works (e.g., Geiping et al., 2025). The paper’s primary contribution is initializing recurrent weights from pretrained models. This is somewhat closer to Bae et. al 2024. The distinctions with respect to these works is fairly minor in my opinion.

**Reviewer Concerns:**

Reviewer 3XYK: This reviewer explicitly characterized the method as an "engineering trick" and noted that copying weights from pretrained models is a technique "existing for a long time." In particular, they questioned the fundamental novelty of the work. While I understand the authors justification that engineering and implementation details are crucial, I think the novelty of the paper is not sufficient for acceptance of the paper in the current form.

Reviewer sNb3: The reviewer had concerns about the presentation and terminology of the paper. I think this concern has been addressed sufficiently in my opinion.

Reviewer pTpx: The review for the paper is not very rigorous and fairly short review.

**Reviewer Scores:**

Reviewer pTpx: Probably will keep their high score but their review is fairly limited.

Reviewer sNb3: Increase their score but still on the borderline.

Reviewer 3XYK: increase their score but still on the borderline.

---

### Decision · Program_Chairs · 2026-01-26

Reject